# Medication adherence trajectories and association with risk factors and clinical outcomes in type 2 diabetes treatment

Sara Mucherino[1,2]*, Valentina Orlando[1,2], Marcia Vervloet[3], Yvette Weesie[3], Liset van Dijk[3,4], Enrica Menditto[1,2]

**1** Department of Pharmacy, University of Naples Federico II, Naples, Italy, **2** Center of Drug Utilization and Pharmacoeconomics (CIRFF), University of Naples Federico II, Naples, Italy, **3** Netherlands Institute for Health Services Research (Nivel), Utrecht, The Netherlands, **4** Department of PharmacoTherapy, Epidemiology & Economics (PTEE), Faculty of Science and Engineering, Groningen Research, Institute of Pharmacy, University of Groningen, Groningen, The Netherlands

* sara.mucherino@unina.it

## Abstract

### Background

Monitoring medication adherence and main clinical outcomes changes in type 2 diabetes (T2D) is essential for optimal patient management. This study aimed to compute and evaluate medication adherence in its three process phases among T2D patients treated with oral antidiabetic (OAD) medications assessing association with main clinical outcomes.

### Methods

This retrospective cohort study included newly diagnosed T2D patients initiating OAD therapy within 2015–2019 identified from the Nivel Primary Care Database (Nivel-PCD) in the Netherlands. Initiation was operationalised as ≥2 OAD prescriptions within follow-up. Implementation was quantified monthly using the Continuous Multiple-interval Measure of Medication Availability (CMA9). Group-based trajectory modelling (k-means via AdhereR) identified adherence patterns over 12 months. Persistence was assessed with a permissible-gap rule on the same 12-month series. Clinical measures (HbA1c, LDL cholesterol, blood pressure, BMI) were evaluated at baseline and at 12 months, and HbA1c trends were analysed over time.

### Results

Among 3,404 T2D patients started an OAD treatment, four distinct adherence clusters were identified: perfect adherence (70.1%), slow decline (13.3%), low adherence (10.6%), and slow increase (6.0%). Overall initiation was 99%. One-year persistence ranged from 26% (low adherence group) to 98.8% (perfect adherence group).

**Data availability statement:** Data of the Nivel Primary Care Database (Nivel-PCD) contain potentially identifying or sensitive patient information. Access to data in Nivel-PCD is subject to Nivel-PCD governance codes. Requests for access to the data can be directed at datarequest@nivel.nl. Restrictions include approval by the appropriate Nivel-PCD governance bodies (privacy committee and steering committee).

**Funding:** This study was supported by: the Italian Medicine Agency (AIFA, Pharmacovigilance funds 2015-2017), E63C23000440002 (received by Prof Valentina Orlando); and PRIN 2022 - European Union (NextGenerationEU), 20227C2YLA (received by Prof Enrica Menditto).

**Competing interests:** The authors declare no conflict of interest. The funders had no role in the design and conduct of the study collection, management, analysis, and interpretation of the data; preparation, review, or approval of the manuscript; and decision to submit the manuscript for publication.

Patients in the perfect-adherence group had higher baseline HbA1c but showed lower HbA1c and LDL levels over time compared with other groups. Blood pressure improved slightly in all groups. Changes in BMI were minimal.

## Conclusions

Adherence trajectories provide a dynamic view of patients' behaviour and are associated with better glycaemic and lipid control. High-risk patients were often in the perfect-adherence group and appeared to gain the highest clinical benefit. These results support routine adherence monitoring and targeted, trajectory-informed interventions in T2D care.

## Introduction

Type 2 diabetes (T2D) is a complex metabolic disorder characterized by chronic hyperglycaemia, and it carries a significant burden of comorbidities, including cardiovascular disease, neuropathy, and retinopathy. It is crucial to address the importance of medication adherence in the context of type 2 diabetes, as it directly affects glycaemic control and, consequently, the progression of associated complications [1]. Poor medication adherence worsens health outcomes and addresses significant public health challenges and economic costs. Clinical management of type 2 diabetes patients include the measurement of a range of health parameters, such as blood cholesterol levels, blood pressure, body weight, and particular attention to glycated haemoglobin (HbA1c) as a key marker of long-term glycaemic control [2]. As already demonstrated, HbA1c measurement provides crucial insights into the effectiveness of diabetes pharmacological management [1]. Indeed, elevated HbA1c levels are associated with an increased risk of T2D-related complications [1]. Beyond glycaemic control, routine T2D management targets LDL cholesterol, blood pressure and body weight [2]. These risk factors are monitored in primary care and drive long-term cardiovascular risk. Therefore, the management of T2D patients should include close monitoring assessing adherence to prescribed medication. This attention would maximise therapeutic effectiveness and support the reduction of key clinical parameters. In fact, in real practice medication adherence is often assessed using a single summary estimate, typically expressed as a percentage. The value measured is used to classify patients as adherent or non-adherent based on a predefined cut-off point or threshold [3,4]. This approach is informative but do not capture the complexity and temporal dynamics of adherence behaviour also not considering variability across patients. According to the EMERGE guidelines and the ABC taxonomy on adherence [5,6], adherence should be evaluated as a process composed of three distinct phases (initiation, implementation and discontinuation/persistence). In this context, applying a longitudinal clustering approach allows to identify subgroups of patients with similar adherence behaviours over time [7–10]. This method is therefore able to capture the different patients' behaviours offering a more accurate understanding of how pharmacological treatment is implemented. It also enables to stratify monitoring

of clinical parameters across adherence subgroups and identify association between adherence and clinical outcomes [11–17]. Using this approach can provide valuable information to healthcare professionals helping in the management of diabetes. Thus, the aim of this study was to compute and evaluate medication adherence in its three process phases among T2D patients treated with oral antidiabetic medications and its association with clinical outcomes.

## Materials and methods

### Data source and study population selection

This is a retrospective longitudinal cohort study. This study was carried out based on data sourced from Nivel Primary Care Database (Nivel-PCD). Nivel-PCD systematically collects data from approximately 500 general practices in the Netherlands (around 10% of the Dutch population) including prescribed medications recorded by general practitioners (GPs) [18]. Hence, Nivel-PCD contains GPs' prescribing records, namely medications ordered by the GPs within the Dutch health system. Pharmacy dispensing/claims data is not captured in Nivel-PCD. In Dutch primary care, long-term therapies are issued as repeat prescriptions with an authorisation typically valid for up to 12 months. Community pharmacies generally dispense 1–3 per month quantities per issue. Insulins and oral antidiabetic drugs (OADs) follow the same framework. Each prescription record in Nivel-PCD includes the calendar date, Anatomical Therapeutic Chemical Classification (ATC) code, prescribed daily dose/instructions, and intended duration or quantity. These fields were used to derive days of coverage per prescription and to build 12-month prescription histories from the index date (first OAD prescription). For this study, these data were integrated with patient characteristics, healthcare consultations, patient morbidity profiles, laboratory results, and the primary or secondary healthcare provider managing diabetes care. In the Netherlands, type 2 diabetes accounts for 82% of all diabetes cases [19], with negative consequences for public health [20–22]. The age and gender distribution of patients registered in Nivel-PCD reflects those of the general Dutch population with type 2 diabetes [23]. Inclusion criteria included all patients first diagnosed with T2D (identified by the International Classification of Primary Care [ICPC] code T90) between 2015 and 2019 within the Nivel PCD data source. Index date was set as the date of the first prescription of an oral antidiabetic (OAD) medication. OADs were identified in the Nivel-PCD by the ATC code A10B, as follows: biguanides (A10BA), sulfonylureas (A10BB and A10BC), combinations of oral blood glucose lowering drugs (A10BD), alpha glucosidase inhibitors (A10BF), thiazolidinediones (A10BG), DPP-4 inhibitors (A10BH), SGLT2 inhibitors (A10BK) and other blood glucose lowering drugs, excluding insulins (A10BX). The OAD prescribed at the index date was defined as the index drug. Medication adherence for each patient in the cohort was calculated with reference to this index drug. Each patient was followed for 12 months from index (one-year follow-up). Patients with any insulin prescription before the index date were excluded. Baseline cohort characteristics were observed within the 90 days prior to the index date for each patient. All baseline covariates were assessed within this window (e.g., age, sex, comorbidity measures, concomitant chronic medicines/diagnoses). For each patient, follow-up started at the index date and continued for 12 months (sliding one year follow-up window per patient). Observation period was censored on 31 December 2020. Patients who discontinued the index OAD prescription before 12 months were not excluded. Their discontinuation contributed to the persistence endpoint, while they continued to contribute on outcome data within the observation window (S1 Fig). Insulin addition was evaluated for each patient and defined as the first insulin prescription initiated after the index date within the follow-up.

### Trajectory modelling and medication adherence estimation

Medication adherence assessment was performed for each patient within the 12-month prescription history (one-year follow-up) starting from the index date. This single time series (days 0–365) was the basis for all adherence measures. Medication adherence was computed according to the EMERGE guidelines [5], distinguishing the three different

phases of the adherence process. Initiation phase was calculated as the occurrence of at least two prescriptions of the index oral antidiabetic drug (OAD) within the one-year follow-up period, since dispensing data were not available. Thus, if a second OAD prescription occurred after the first (index) OAD prescription, the patient was classified as an initiator, and the index date was considered as the initiation date. Persistence phase was computerised from prescribing patterns of OADs, considering the prescribed coverage period of each individual OAD. Patients were classed as non-persistent if no later OAD prescription was recorded, or if the gap to the next prescription was > 2 × the expected days' supply of the previous one. This approach identifies gaps in treatment that indicate discontinuation during the follow-up period. If a patient switched to another OAD and no further doses of the reference drug were prescribed, the reference therapy was considered discontinued at the first permitted break in the therapeutic interval. When another OAD was added and prescriptions for the index drug continued, the index trajectory was unchanged. Patients with only one OAD prescription during follow-up were reported under initiation but excluded from persistence and implementation estimations as, according to the EMERGE guidelines, persistence is time-to-event. This is because these two phases require ≥2 prescriptions to observe continued exposure and to avoid misclassification in the trajectories. Indeed, as persistence calculation is time-to-event, it is not possible to determine whether the only one OAD prescription was taken by the patient or not (e.g., the patient does not start the prescribed treatment). Implementation phase was calculated and plotted by using the trajectory-based modelling, mainly based on clustering approach. Clustering method used consisted of a set of multivariate techniques aimed at recognising and grouping homogeneous items in a data set by using non-parametric statistics [10,24–26]. The clustering approach grouped patients with similar month-to-month adherence rates over the first year of the OAD-therapy. Each cluster was summarized by a centroid, i.e., the typical adherence curve formed by averaging the profiles of its members. Then, the clusters were labelled by the shape of their centroid to reflect recognizable behaviours in adherence. Hence, cluster analysis for adherence implementation measurement allowed the identification of groups of T2D patients with common characteristics (e.g., potential determinants or predictors of risk) [14,24–27]. This approach linked heterogeneous individual adherence patterns into a small set of clinically interpretable behaviours. The indicator used for the adherence calculation in its implementation was the continuous multiple interval measure of medication availability/gaps (CMA) version 9 [28–30]. For the whole observation period, the CMA9 was computed as the ratio of theoretical medication use days to the adherence assessment period duration, accounting for prescriptions carryover and excluding end-period prescriptions. CMA9 provides an average adherence measure over time. It weights days by each prescription's coverage and calculates adherence in non-overlapping one-month windows, aligned with the typical interval between successive prescriptions. Hence, prescriptions histories for each index OAD over an observation period one year were calculated. In Dutch primary care, chronic therapies are authorised as repeat prescriptions by the GP, commonly for up to 12 months. Community pharmacies then dispense smaller quantities according to standard practice (often 1–3 months at a time for both insulins and OADs [27,31]). Nivel-PCD records the GP prescription authorisations and dates. Hence, medication coverage was computed using successive prescription records. Specifically, days of supply were derived at the prescription level from the recorded prescription duration (days) and the prescribed daily dose. When this field was unavailable or inconsistent, days of supply were inferred from molecule/formulation-specific dosing (national guidance [27]) and the spacing between successive prescriptions. Coverage was carried over across prescriptions and summarized in one-month sliding windows for CMA9. Then, the actual doses prescribed (dosage and coverage) were compared with the doses recommended by the NHG guideline for diabetes mellitus type 2 [27] and the Medicines Information Bank managed by the Dutch Medicines Evaluation Board (MEB) [31]. NHG recommended posology related to OADs are: once to twice daily for sulfonylureas, thiazolidinediones, DPP-4 inhibitors and SGLT2 inhibitors; 1–3 times a day for combinations of oral blood glucose lowering drugs, alpha glucosidase inhibitors, other blood glucose lowering drugs (excluding insulins); 2–3 times a day for biguanides considering that the extended-release tablet is taken once daily. Therefore, the posology of all OAD medications was settled ad-hoc per each medicinal speciality.

                                                                    

## Model covariates

Baseline characteristics and post-index events, such as insulin initiation, were analysed for each adherence group. Patients' complexity was assessed calculating: i) rates of multimorbidity (two or more chronic conditions at baseline based on ICPC codes) and polypharmacy (concurrent use of five or more different medications at baseline based on ATC codes); ii) index of high risk T2D patients according to the Dutch treatment and diagnostic guidelines for T2D [27]: patient who also suffer or suffered from angina pectoris, acute myocardial infarction, other and chronic ischemic heart disease, heart failure, stroke/cerebrovascular accident, atherosclerosis, other arterial obstruction, vascular disease, chronic alcohol abuse, obesity (BMI > 30) and symptoms/complaints kidney; iii) Charlson comorbidity (CCI) score [26]: the score assigns weights for a number of major conditions included in secondary diagnoses. The CCI score appears as the total value of the assigned weights and pertaining measure of the weight of each comorbidity. CCI scores were categorized into three distinct grades: low, characterized by CCI scores of 0–1; mild, corresponding to CCI scores of 2–3; and severe, indicated by CCI scores of ≥4.

## Clinical outcomes

Association between medication adherence and main T2D clinical outcomes was investigated in accordance with the guideline of the Dutch College of General Practitioners (NHG) for T2D monitoring [27]. The guideline suggests that T2D patients must undergo quarterly check-ups aimed at evaluating potential diabetes-related complications. These check-ups include the assessment of specific clinical parameters against clinical target levels according to WHO guidelines. The main clinical parameter includes haemoglobin A1c levels (HbA1c; clinical target ≤53 mmol/mol/ ≤7%). Also, secondary outcomes are LDL cholesterol levels (target <2.8 mmol/L), body weight measured through Body Mass Index (BMI; target <27), and blood pressure (BP; target <140/90 mmHg). These latter were treated as secondary, descriptive outcomes. Hence, clinical parameters were observed for each T2D patient within 90 days before the index date and within 90 days after the one-year follow-up period (S1 Fig). Thus, the end of the entire study period was on 31 March 2021. If the T2D patient had several clinical parameter measurements before the index prescription or after the end of follow-up, the measurement closest in time to these two set dates was considered. Additionally, number of clinical measurements collected for each T2D patient within the one-year follow-up period was also retrieved to assess temporal variation and compare trends across adherence groups.

## Statistical analyses

Covariates were described using the mean and standard deviation (SD) or median and interquartile range (IQR) for continuous variables, and frequency and percentage for categorical variables. Adherence trajectory groups were treated as the exposure accumulated during the one-year follow-up. Pre/post changes in clinical parameters by adherence trajectory were compared. Because adherence trajectories and outcomes are derived over the same 12-month horizon, all contrasts were interpreted as associations rather than causal effects. Clustering in adherence groups was performed with the AdhereR package (version 0.8.10) and the R package "kml" (version 2.4.1), which provided an implementation of k-means designed to work specifically on longitudinal data [28–30]. The number of adherence groups was determined by comparing model fit statistics across solutions ranging from two to six groups of adherence (k = 2–6) using one-year CMA9 series. Implementation exposure was constructed as monthly CMA9 values over 12 months (rolling 1-month windows) [24–26]. Solutions with a multi-criteria panel of internal indices (standardized Calinski–Harabasz CH- CH2- CH3, Ray-Turi, Davies-Bouldin) were compared. Cluster stability was assessed in three steps. First, k-means was run with 100 random initialisations to reduce dependence on starting values. Second, bootstrap resampling was applied (200 samples with replacement). For each bootstrap sample, the solution was compared with the full-sample solution using the Jaccard index for cluster membership. Specifically, values ≥0.75 were considered stable and values 0.60–0.75 moderately stable.

Third, a split-sample check was performed: centroids were estimated in a random 50% training subset, and the remaining 50% were assigned to the nearest centroid. Finally, agreement with the full-sample assignment was then evaluated. Moreover, sensitivity analyses were performed as follow: 1) the monthly CMA9 construction (±3-day tolerance around month boundaries); 2) application of a light 3-point moving average (yes/no); 3) a minimum cluster size of 5% to avoid small clusters. Finally, the selection rule was the smallest k with jointly favourable indices, stable membership (Jaccard ≥ 0.75), and distinct (also clinically interpretable) trajectory shapes [24–26].

Multivariate logistic regression was performed to identify factors associated with adherence clusters. A stepwise selection procedure was used to select covariates to be included in the models. These covariates included: sex, age category (<25, 26–39, 40–59, 60–79, ≥ 80 years), high-risk T2D status per NHG guidance [27], hypertension at baseline, baseline HbA1c within target (≤53 mmol/mol; 7%), baseline blood pressure within target (<140/90 mmHg), and insulin addition during follow-up (post-index indicator). Results are presented as adjusted odds ratios (aORs) with 95% confidence intervals and two-sided p-values (α = 0.05). Linear mixed-effect regression model was used to model HbA1c over the 12-month follow-up and its association with adherence groups. All HbA1c measurements recorded during follow-up were included. The fixed-effects structure comprised time since index (measured in months), adherence trajectory group, and their interaction (time x group), with baseline HbA1c entered as an adjustment covariate. Random effects included patient-level intercepts and slopes for time, to consider the association within-patient and heterogeneous trajectories. Model coefficients, interaction contrasts, and monthly slopes of HbA1c were reported with 95% confidence intervals. For both models p value <0.05 was considered statistically significant.

Data management was performed with Stata, Statistical software for data science. Statistical analyses were performed with R (version 3.6, The R Formulation for Statistical Computing).

## Results

### Adherence groups and cohort characteristics

The overall cohort consisted of 3,404 patients first diagnosed with T2D. A flow chart reporting final cohort selection is showed in S2 Fig. Baseline cohort characteristics assessed at the index date are detailed in Table 1 and S1 Table. Regarding the clustering, the solution of k = 4 clusters achieved the near-highest standardized CH indices and favourable Davies-Bouldin. The bootstrap stability for the k = 4 clusters solution was high (cluster-wise Jaccard 0.79–0.88; overall mean 0.84) and split-sample agreement was 89%. Sensitivity analyses produced highly correlated centroids (r = 0.93–0.98) and similar group proportions (absolute differences ≤ 1.8%) (see S3 and S4 Figs). The identified four adherence groups were as follows:

i)  Group A: Perfect Adherence n=2,386 T2D patients (70.1%).

ii)  Group B: Slow decline in adherence n = 453 T2D patients (13.3%).

iii)  Group C: Low Adherence n=362 T2D patients (10.6%).

iv)  Group D: Slow increase in adherence n = 203 T2D patients (6.0%).

Overall, the initiation phase analysis revealed that a small proportion of T2D patients of the cohort (0.9%) had a single OAD prescription. However, Group C (low adherence) had the highest proportion of patients with only one OAD prescription (6.9%) (S2 Table). In the persistence phase, using the pre-specified permissible-gap definition, 84.6% of the overall T2D cohort remained on therapy at the end of the one-year follow-up. Specifically, patients in group A (perfect adherence) had the highest persistence rate (98.8%) at the end of the follow-up period, followed by group D (slow increase in adherence, 72.4%), group B (slow decline in adherence, 62%) and group C (low adherence, 26%) (S2 Table).

Regarding the implementation phase, the mean CMA value across the total cohort was 0.8 (SD 0.3) with differences between groups: group A maintained optimal implementation (CMA 1.0), group C recorded very poor implementation (CMA

**Table 1. Patients' baseline characteristics by each adherence group.**

| Adherence Groups | Group A Perfect Adherence | Group B Slow decline | Group C Low Adherence | Group D Slow increase |
|---|---|---|---|---|
| **T2D Cohort**, n (%) | **2,386 (70.1)** | **453 (13.3)** | **362 (10.6)** | **203 (6.0)** |
| Males, n (%) | 1,470 (61.6) | 288 (63.6) | 201 (55.5) | 123 (60.6) |
| Females, n (%) | 916 (38.4) | 165 (36.4) | 161 (44.5) | 80 (39.4) |
| **Age, mean (SD)** | 60 (12.9) | 58.7 (13.6) | 61.2 (14.7) | 58.8 (13.8) |
| Aged under 25 years, (%) | 8 (0.3) | 7 (1.5) | 1 (0.3) | 3 (1.5) |
| Aged 26–39 years, (%) | 109 (4.6) | 28 (6.2) | 28 (7.7) | 14 (6.9) |
| Aged 40–59 years, (%) | 1,085 (45.5) | 202 (44.6) | 129 (35.6) | 83 (40.9) |
| Aged 60–79 years, (%) | 998 (41.8) | 185 (40.8) | 163 (45.0) | 93 (45.8) |
| Aged over 80 years, (%) | 186 (7.8) | 31 (6.8) | 41 (11.3) | 10 (4.9) |
| OAD treatment, n (%) **Biguanides** | 2,226 (93.3) | 422 (93.2) | 319 (88.1) | 182 (89.7) |
| OAD treatment, n (%) **Sulfonylureas** | 151 (6.3) | 30 (6.6) | 37 (10.2) | 19 (9.4) |
| OAD treatment, n (%) **Combinations of oral blood glucose lowering drugs** | 2 (0.1) | 1 (0.2) | 2 (0.6) | – |
| OAD treatment, n (%) **Alpha glucosidase inhibitors** | 1 (<0.1) | – | – | – |
| OAD treatment, n (%) **Thiazolidinediones** | 1 (<0.1) | – | – | 1 (0.5) |
| OAD treatment, n (%) **DPP-4 inhibitors** | 4 (0.2) | – | 2 (0.6) | 1 (0.5) |
| OAD treatment, n (%) **SGLT2 inhibitors** | – | – | 1 (0.3) | – |
| OAD treatment, n (%) **Other blood glucose lowering drugs, excl. insulins** | 1 (<0.1) | – | 1 (0.3) | – |
| **High risk type 2 diabetes patients**, n (%) | **298 (12.5)** | **42 (9.3)** | **33 (9.1)** | **23 (11.3)** |
| Angina pectoris | 53 (17.8) | 3 (7.1) | 6 (18.2) | 2 (8.7) |
| Acute myocardial infarction | 78 (26.2) | 12 (28.6) | 10 (30.3) | 5 (21.7) |
| Other and chronic ischaemic heart disease | 46 (15.4) | 6 (14.3) | 4 (12.1) | 5 (21.7) |
| Heart failure | 51 (17.1) | 5 (11.9) | 4 (12.1) | 2 (8.7) |
| Stroke/cerebrovascular accident | 51 (17.1) | 9 (21.4) | 3 (9.1) | 6 (26.1) |
| Atherosclerosis | 12 (4.0) | 2 (4.8) | 1 (3.0) | 1 (4.3) |
| Other arterial obstruction/pheriph. vascular disease | 22 (7.4) | 5 (11.9) | 5 (15.2) | 1 (4.3) |
| Chronic alcohol abuse | 2 (0.7) | – | – | – |
| Obesity (BMI > 30) | 8 (2.7) | 1 (2.4) | 1 (3.0) | 3 (13.0) |
| Symptoms/complaints kidney | 3 (1.0) | 1 (2.4) | 1 (3.0) | – |
| **Charlson comorbidity score**, mean (SD) | 2.0 (10.3) | 1.5 (6.9) | 2.0 (7.6) | 2.6 (11.2) |
| CCI Low score (0–1), n (%) | 2,102 (88.1) | 399 (88.1) | 312 (86.2) | 171 (84.2) |
| CCI Mild score (2–3), n (%) | 61 (2.6) | 12 (2.6) | 6 (1.7) | 6 (3.0) |
| CCI Severe score (≥4), n (%) | 223 (9.3) | 42 (9.3) | 44 (12.2) | 26 (12.8) |
| **Comorbid conditions**, mean (SD) | 2.4 (2.2) | 2.2 (2.2) | 2.2 (2.3) | 2.9 (2.5) |
| **Comorbid conditions**, median number (IQR) | 2 (1-3) | 2 (1-3) | 2 (1-3) | 2 (1-4) |
| **Prescriptions per year,** mean nr (SD) | 39.5 (59.7) | 22.1 (27.3) | 20.2 (29.7) | 29.0 (47.3) |
| **Prescriptions per year,** median number (IQR) | 22 (13-36) | 15 (9-27) | 12 (6-25) | 19 (13-32) |
| Concomitant medication, n (%) **Lipid Modifying Agents (C10)** | 1361 (57.0) | 201 (44.4) | 139 (38.4) | 126 (62.1) |

*(Continued)*

**Table 1.** (Continued)

| Adherence Groups | Group A Perfect Adherence | Group B Slow decline | Group C Low Adherence | Group D Slow increase |
|---|---|---|---|---|
| Concomitant medication, *n (%)* **Drugs for Acid Related Disorders (A02)** | 956 (40.1) | 178 (39.3) | 139 (38.4) | 85 (41.9) |
| Concomitant medication, *n (%)* **Agents acting on Renin-Angiotensin System (C09)** | 899 (37.7) | 132 (29.1) | 104 (28.7) | 76 (37.4) |
| Concomitant medication, *n (%)* **Antithrombotic Agents (B01)** | 727 (30.5) | 125 (27.6) | 93 (25.7) | 61 (30.0) |
| Concomitant medication, *n (%)* **Beta Blocking Agents (C07)** | 649 (27.2) | 98 (21.6) | 82 (22.7) | 63 (31.0) |
| Concomitant medication, *n (%)* **Diuretics (C03)** | 595 (24.9) | 84 (18.5) | 76 (21.0) | 53 (26.1) |
| Concomitant medication, *n (%)* **Psycholeptics (N05)** | 386 (16.2) | 84 (18.5) | 65 (18.0) | 43 (21.2) |
| Concomitant medication, *n (%)* **Calcium Channel Blockers (C08)** | 407 (17.1) | 62 (13.7) | 40 (11.0) | 36 (17.7) |
| Concomitant medication, *n (%)* **Drugs for Obstructive Airway Diseases (R03)** | 358 (15.0) | 64 (14.1) | 53 (14.6) | 34 (16.7) |
| Concomitant medication, *n (%)* **Psychoanaleptics (N06)** | 335 (14.0) | 55 (12.1) | 38 (10.5) | 29 (14.3) |
| Other chronic comorbidity, *n (%)* **Hypertension** | 555 (23.3) | 78 (17.2) | 61 (16.9) | 56 (27.6) |
| Other chronic comorbidity, *n (%)* **Lipid metabolism disorder** | 197 (8.3) | 37 (8.2) | 19 (5.2) | 17 (8.4) |
| Other chronic comorbidity, *n (%)* **Asthma** | 97 (4.1) | 20 (4.4) | 17 (4.7) | 14 (6.9) |
| Other chronic comorbidity, *n (%)* **COPD** | 97 (4.1) | 16 (3.5) | 13 (3.6) | 9 (4.4) |
| Other chronic comorbidity, *n (%)* **Atrial fibrillation/flutter** | 84 (3.5) | 12 (2.6) | 8 (2.2) | 8 (3.9) |
| Other chronic comorbidity, *n (%)* **Depressive disorder** | 76 (3.2) | 9 (2.0) | 11 (3.0) | 7 (3.4) |
| Other chronic comorbidity, *n (%)* **Gout** | 72 (3.0) | 10 (2.2) | 10 (2.8) | 7 (3.4) |
| Other chronic comorbidity, *n (%)* **Hypothyroidism/myxoedema** | 68 (2.8) | 10 (2.2) | 10 (2.8) | 7 (3.4) |

mean 0.3, SD 0.1) and group B and D had intermediate profiles (CMA mean 0.6 and 0.7 respectively, with SD 0.1) (S2 Table). Age differences were identified across the four adherence groups (Table 1). Mainly, about half of patients in the perfect adherence group and slow decline in adherence group were aged between 40–59 years (group A: n = 1,085, 45.5%; group B: n = 202, 44.6%) while half of the patients in the low adherence group and slow increase in adherence group were aged between 60–79 years (group C: n = 163, 45%; group D: n = 93, 45.8%). Moreover, 17.4% of patients with low adherence to OADs treatment and 12.1% of patients with slow decline in adherence added an insulin treatment 2½ months after the start of OADs treatment (S3 Table). The highest proportion of high risk T2D patients was identified in the perfect adherence group (group A, 12.5%). Of these, 26.2% suffered from an acute myocardial infarction, and in the slow increase in adherence group (group D, 11.3%), of these 26.1% had a stroke/cerebrovascular event. Patients in the low adherence group C recorded the highest comorbidity score (mean: 2.6; SD: 11.2). Regarding the overall T2D cohort multimorbidity level, the mean number of comorbid conditions was higher for patients in the slow increase in adherence group (group D) with about three additional chronic conditions to T2D.

## Adherence groups and clinical outcomes

Variation in clinical outcomes from the index date to the end of the one-year follow-up recorded differences between the four adherence groups (Table 2, S5 and S6 Figs). The clinical parameters target values to be monitored in patients with T2D are described in S4 Table.

Prior to the start of OADs treatment (index date), 70.5% of T2D patients with perfect adherence (group A) had measured HbA1c levels >53 mmol/mol, but after one year from the start of therapy (end of follow-up) only 31.5% of these were still above the clinical target. Highest variation was also observed for the group D (slow increase in adherence) recording 61.6% of patients above the HbA1c clinical target at the beginning of follow-up and 25.1% at the end (Table 2, S5 and S6 Figs). Moreover, HbA1c levels at the end of follow-up were measured below the target value for all adherence groups except for the patients in the low adherence (group C 53.7 mmol/mol at the end of follow-up) (see Fig 1 and S5 Table) although they started from above target levels at baseline (S7 Fig). Additionally, the mixed-effects model showed that all groups had a negative monthly slope in HbA1c after OAD initiation (overall −2.93 mmol/mol per month). Compared with the perfect adherence group A, the slope was significantly less steep in group B (difference +0.44 mmol/mol per month; 95% CI 0.08–0.80; p = 0.02) and group C (+1.07; 95% CI 0.61–1.52; p < 0.001), indicating slower HbA1c improvement in these groups (Fig 1 and S6 and S7 Tables).

Normal LDL levels (<2.8 mmol/L) were only reached by T2D patients in the perfect adherence group (group A). Albeit this, at the end of the follow-up, 38.1% of perfect adherent were still above the LDL clinical target. Regarding BMI

**Table 2. Variation in outcome measures from the beginning to the end of follow-up.**

| Adherence Groups | Type 2 diabetes Cohort | Group A Perfect Adherence | Group B Slow decline | Group C Low Adherence | Group D Slow increase |
|---|---|---|---|---|---|
| **T2D Cohort**, n | **3,404** | **2,386** | **453** | **362** | **203** |
| **T2D Cohort**, % | **100%** | **70.1%** | **13.3%** | **10.6%** | **6.0%** |
| **N of measurements of HbA1c in 1year**, mean (SD) | 6 (4.2) | 6.2 (4.1) | 5.3 (3.5) | 4.1 (3) | 6.1 (5.6) |
| **HbA1c > 53 mmol/mol(index date)**, n (%) | 2,225 (65.4) | 1,683 (70.5) | 257 (56.7) | 160 (44.2) | 125 (61.6) |
| **HbA1c mean value(index date)**, mmol/mol (%) | 70.2 (8.6) | 72 (8.7) | 65.3 (8.1) | 63.8 (8.0) | 67.9 (8.4) |
| **HbA1c > 53 mmol/mol (end of FU)**, n(%) | 1,022 (30) | 752 (31.5) | 122 (26.9) | 97 (26.8) | 51 (25.1) |
| **HbA1c mean value(end of FU)**, mmol/mol (%) | 50.3 (6.7) | 50.3 (6.7) | 49.1 (6.6) | 53.7 (7.1) | 47.9 (6.5) |
| **N of LDL measurements in 1year**, mean (SD) | 3.2 (2.6) | 3.6 (2.5) | 3 (2.3) | 3 (2.6) | 3.5 (2.5) |
| **LDL > 2.8 mmol/L (in ex date)**, n(%) | 1,850 (54.3) | 1,340 (56.2) | 237 (52.3) | 165 (45.6) | 108 (53.2) |
| **LDL mean value (index date)**, mmol/L | 3.2 | 3.2 | 3.2 | 3.1 | 3.2 |
| **LDL > 2.8 mmol/L (end of FU)**, n (%) | 1,307 (38.4) | 910 (38.1) | 185 (40.8) | 141 (39) | 71 (35) |
| **LDL mean value (end of FU)**, mmol/L | 2.8 | 2.7 | 2.9 | 2.9 | 2.8 |
| **N of BMI measurements in 1year**, mean (SD) | 5.4 (3.3) | 4.6 (3.2) | 5.1 (4.3) | 3.7 (2.3) | 5.4 (4.5) |
| **BMI > 27 (index date)**, n (%) | 2,022 (59.4) | 1,489 (62.4) | 246 (54.3) | 167 (46.1) | 120 (59.1) |
| **BMI mean value (index date)**, mean value | 30.7 | 30.7 | 30.5 | 30.6 | 31.1 |
| **BMI > 27 at the end of FU**, n (%) | 1,907 (56) | 1,406 (58.9) | 227 (50.1) | 161 (44.5) | 113 (55.7) |
| **BMI mean value (end of FU)**, mean value | 30.1 | 30.2 | 29.8 | 30 | 30.4 |
| **N of BP measurements in 1year**, mean (SD) | 6.1 (4.5) | 6.8 (4.2) | 5.8 (6.0) | 4.1 (3.6) | 6.1 (5.7) |
| **BP > 140/90 mmHg (index date)**, n(%) | 1,268 (37.3) | 944 (39.6) | 145 (32) | 112 (30.9) | 67 (33) |
| **BP mean value (index date)**, mmHg | 140/83 | 140/84 | 138/82 | 139/82 | 137/82 |
| **BP > 140/90 mmHg (end of FU)**, n(%) | 819 (24.1) | 607 (25.4) | 95 (21) | 72 (19.9) | 45 (22.2) |
| **BP mean value (end of FU)**, mmHg | 133/80 | 133/81 | 133/79 | 133/79 | 133/79 |

Abbreviations: BMI, Body Mass Index; BP, Blood pressure; FU, Follow up; HbA1c, Haemoglobin A1c; LDL, Low-density lipoprotein; SD, Standard deviation.

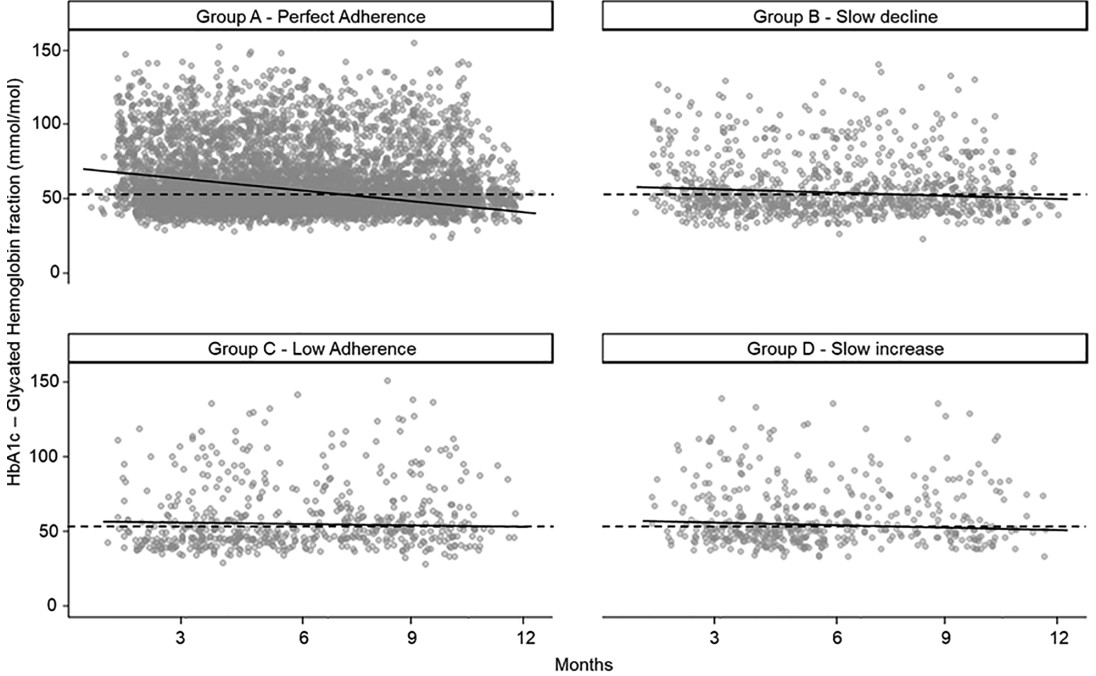

**Fig 1. Linear mixed-effect regression model correlating changes in HbA1c levels and adherence groups.** The regression line reflects the mixed-effects model that uses all HbA1c timestamps and includes time since index date and baseline HbA1c. Each grey dot represents an individual HbA1c measurement recorded during the follow-up for a T2D patient. The black line illustrates the fitted linear mixed-effects trend. The model includes time since index (months), adherence group, and their interaction, adjusts for baseline HbA1c. The model also uses patient-level random intercepts and slopes. The dashed horizontal line represents the clinical HbA1c target threshold, set at 53 mmol/mol (7%).

parameter, most T2D patients across all adherence groups were overweight at baseline (BMI > 27), especially in the perfect adherence (group A: 62.4%) and slow increase (group D: 59.1%) groups. At the end of follow-up, mean BMI values recorded slightly lower levels in all adherence groups (e.g., from 30.7 to 30.2 in group A).

Blood pressure control recorded lower levels for all the adherence groups from the baseline to the end of follow-up. The proportion of T2D patients with BP > 140/90 mmHg declined in all groups, from 39.6% to 25.4% in the perfect adherence group A. Mean BP values reached similar levels across groups at the end of follow-up (approximately 133/80 mmHg).

Logistic regression models were performed to evaluate the association between determinant variables with the adherence group, as shown in Table 3. Perfect adherence behaviour to OADs treatment was associated with age 40−59 years (OR 2.83, CI 95% 1.12–7.18) diagnosis of hypertension (OR 1.21, CI 95% 1.00–1.47) and with a high-risk T2D (OR 1.35, CI 95% 1.05–1.73). Additionally, perfect adherence was associated to subjects reaching normal values of HbA1c after one-year OADs treatment (OR 1.35, CI 95% 1.14–1.60) as for those adding insulin to the initial OAD treatment after the index date (OR 0.55, CI 95% 0.42–0.71).

The model also revealed that patients with hypertension and those with normal HbA1c values at the end of follow-up were less likely to belong to the low adherence group (OR 0.7, CI 95% 0.52–0.94 and OR 0.75, CI 95% 0.58–0.96, respectively) and to the slow decline in adherence group (OR 0.76, CI 95% 0.58–0.99 and OR 0.81, CI 95% 0.65–1.02, respectively).

## Discussion

This study mapped medication adherence across its three phases (initiation, implementation, and persistence/discontinuation) during the first year of treatment in newly diagnosed T2D receiving OADs. The three adherence phases were

**Table 3. Multivariate logistic regression analysis to identify determinants of adherence groups.**

| Adherence Groups | Group A Perfect Adherence | | Group B Slow decline | | Group C Low Adherence | | Group D Slow increase | |
|---|---|---|---|---|---|---|---|---|
| Determinant | P | aOR 95% CI | P | aOR 95% CI | P | aOR 95% CI | P | aOR 95% CI |
| Female | 0.51 | 0.95(0.81-1.11) | 0.29 | 0.89(0.72-1.10) | **0.04** | 1.27(1.02-1.59) | 0.97 | 1.00(0.75-1.35) |
| 26-39y* | 0.22 | 1.85(0.70-4.87) | **0.03** | 0.32(0.12-0.90) | 0.16 | 4.47(0.56-3.37) | 0.28 | 0.48(0.12-1.85) |
| 40-59y* | **0.03** | 2.83(1.12-7.18) | **0.01** | 0.29(0.12-0.76) | 0.35 | 2.63(0.34-2.21) | 0.07 | 0.31(0.09-1.11) |
| 60-79y* | 0.07 | 2.35(0.91-4.87) | **0.01** | 0.29(0.11-0.76) | 0.20 | 3.82(0.50-2.32) | 0.11 | 0.36(0.10-1.27) |
| ≥80y* | 0.09 | 2.30(0.88-6.01) | **0.01** | 0.27(0.10-0.74) | 0.12 | 5.20(0.66-4.70) | **0.03** | 0.21(0.05-0.84) |
| High-risk patients | **0.02** | 1.35(1.05-1.73) | 0.20 | 0.80(0.56-1.13) | **0.05** | 0.68(0.46-1.00) | 0.89 | 0.97(0.61-1.53) |
| HTN | **0.05** | 1.21(1.00-1.47) | **0.04** | 0.76(0.58-0.99) | **0.02** | 0.70(0.52-0.94) | **0.03** | 1.43(1.03-1.99) |
| Normal HbA1c values | **<.001** | 1.35(1.14-1.60) | 0.07 | 0.81(0.65-1.02) | **0.02** | 0.75(0.58-0.96) | 0.19 | 0.80(0.58-1.12) |
| Normal BP values | **0.005** | 1.28(1.08-1.53) | **0.04** | 0.78(0.61-0.99) | 0.27 | 0.86(0.66-1.12) | 0.24 | 0.82(0.58-1.14) |
| Insulin addition | **<.001** | 0.55(0.42-0.71) | 0.40 | 1.16(0.81-1.67) | 0.34 | 2.49(0.79-3.45) | 0.78 | 1.08(0.64-1.82) |

Abbreviations: aOR, adjusted odds ratio; 95% CI, 95% Confidence interval; HTN, Hypertension; Y, Age (years); P, P-value *Reference age group: <25 years. Notes: P-values in bold indicate statistical significance (p<0.05).

evaluated in accordance with the most recent adherence guidelines (the EMERGE guidelines [5] and the ABC taxonomy [6]). This study investigated the actual OAD implementation during the first year of treatment. Findings revealed four sub-populations (groups) reflecting different adherence behaviours: patients with perfect adherence during the 1-year treatment; adherent patients but with a slow decline in adherence during the year; patients with low adherence levels during the 1-year treatment; and patients with low adherence levels but with a slow increase in adherence during the year. Higher-adherence patterns were associated with higher risk-factor control over time, particularly lower HbA1c and LDL cholesterol. The OAD-therapy initiation was high (99%), and one-year persistence was consistent with prior T2D reports (70–80%) when considered across adherence subgroups, with marked variation between behaviours [32–34]. These findings indicate that high-risk patients, such as those with more severe disease profiles or multiple comorbidities, are among those with perfect adherence behaviour. Hence, this could confirm, on a real-world sample of diabetic patients, that adherence is significantly influenced by disease severity and comorbidities [35,36]. Specifically, our results showed that, at baseline, T2D patients within the perfect adherence group had the highest HbA1c levels, well above the clinical target. However, during follow-up they showed the more increased control in clinical outcomes, especially in HbA1c and LDL cholesterol. This aspect supports previous studies linking better adherence levels to antidiabetic medications with reductions in HbA1c levels [37,38]. The normalisation of clinical parameters in highly adherent patients highlights the importance of respecting the treatment as prescribed to manage T2D condition and prevent clinical complications. Other studies also support this finding, showing that adherence can be associated to a better glycaemic control which reduces diabetes-related risks [37,38]. The reasons explaining this phenomenon could be several. Firstly, patients with perfect adherence may be more aware of their health and more motivated to follow their treatment. Secondly, they may also be monitored more closely by healthcare providers (e.g., additional follow-up, caregiver involvement, patient empowerment and engagement) [39,40].

Regarding patients' age, the observed differences in age distribution across adherence groups suggest that younger patients (aged 40–59) are more likely to maintain higher levels of adherence. In contrast, older patients (aged 60–79) showed lower or more variable adherence levels. This finding contrasts with previous studies reporting higher adherence rates among older adults [41]. This may be partly explained by differences in study populations and healthcare system setting. Hence, variations in healthcare system structures, care continuity, and medication management across settings may contribute to differences in observed adherence patterns [42]. This variation may indicate that age-related factors (e.g., comorbidities, cognitive decline, or treatment complexity) could have a significant role in

influencing adherence behaviour in older populations [43,44]. This highlights the need for age-tailored interventions to improve adherence in these groups [43,44]. However, other relevant factors not included in our analysis may act as confounders. As recognized by the WHO [42] and widely reported in the literature on adherence research so far [45], determinants such as socioeconomic status, patient related factors (e.g., education and health literacy) may also influence medication-taking behaviour [39,40].

Previous studies have already assessed persistence in T2D patients, but without differentiating adherence levels within its phases [32–34]. Differently to our study, this common approach relies on an overall unique adherence estimate [32–34]. Indeed, previous studies report one-year persistence rates around 70–80%, not considering differences in behaviour among patient subgroups [32–34]. Corroborating this, our findings identified 70% of the T2D cohort within the adherence group, hence reflecting the literature so far. A key strength of this study is going beyond overall adherence rates. The three adherence phases were examined, and four distinct trajectories were identified in T2D therapy. These trajectories were then compared observing the changes in main clinical parameter, namely HbA1c. While only HbA1c was modelled longitudinally, routinely monitored cardiometabolic risk factors (namely LDL, BP and BMI) were summarized descriptively to contextualize cardiometabolic control across adherence patterns. Hence, the four adherence groups identified showed wide variation in persistence, from 26% in low-adherence patients to 98.8% in those with perfect adherence. This includes dynamic behaviours over time, such as increasing (72.4%) or decreasing (62%) adherence levels. These changes may reflect side effects, lack of efficacy, or treatment adjustments. Albeit this, few most recent studies on adherence on T2D, also used a dynamic group-based approach to capture difference adherence behaviours (adherence trajectories) [46–50]. This study is in line with these also supporting methodological advances of the current scenario of medication adherence research, particularly in the context of the diabetic chronic condition. Specifically, comparing to these recent studies, we used the continuous multiple-interval measure of medication availability (CMA) to compute and estimate adherence clusters and persistence rates, rather than the more commonly used proportion of days covered (PDC) [41,46–48]. CMA allows for a more dynamic measurement of adherence over time, especially useful for identifying patterns across different patient groups [28–30]. Second, earlier studies mostly linked trajectories to HbA1c only; to our knowledge, none systematically examined multiple outcomes (e.g., LDL cholesterol, blood pressure, and BMI) by adherence pattern. This represents one of the key strengths of this work, as it provides a broader understanding of how different adherence behaviours influence key clinical outcomes in real-world settings. Moreover, this study supports the use of trajectory modelling to delineate distinct adherence patterns. This modelling approach can be a useful tool to understand adherence dynamics over-time overcoming the limitations of unique adherence measurement [11–18,49–51]. This was also demonstrated by recent longitudinal research using the same approach in different chronic condition [12,36,52,53]. Hence, similarly, Librero et al. (2016) [36] investigated medication adherence patterns after hospitalization for coronary heart disease identifying five adherence patterns. They also recognised the association between older age and adherence groups. Additionally, An et al. (2021) [52] studied long-term medication adherence trajectories to direct oral anticoagulants in patients with atrial fibrillation. They founded that early discontinuation of these medications was associated with a higher risk of thromboembolic events, addressing the clinical importance of adherence. These studies, in line with our results, emphasize the association between dynamic adherence behaviour and clinical outcomes. This aspect highlights the need for continuous adherence monitoring and targeted interventions to better control long-term health outcomes [12,36,53]. Future research should confirm that patients with higher adherence to pharmacological therapies have fewer hospitalizations and emergency department visits. This suggests potential healthcare cost savings as well as more controlled clinical outcomes, as already demonstrated [54]. Nevertheless, this study has several limitations that should be considered. The analyses were based on a data source which contains prescription records of the whole Dutch population. However, prescription records cannot confirm whether patients took or not their prescribed medication. To address this, we considered at least two prescriptions within the observation period as a proxy for treatment initiation and continuation. Hence, days of supply were built on OADs-prescriptions. When not specified within the dataset, medication

coverage was inferred from the dosage recommended by the NHG guidelines [27]. However, the CMA9 monthly windows likely mitigated this error. Data source used also lacks patient-level social determinants (such as socioeconomic status, education, health literacy). Future linkage with external data, such as deprivation indices, would strengthen control for the factors analysed in this study. Moreover, although insulin addition was evaluated as a potential inefficacy of the OAD treatment, the mean time to insulin addition was relatively short in our cohort. Therefore, the decision to initiate insulin treatment may not fully reflect the glycaemic effect of OAD therapy, especially considering that HbA1c reflects glycaemic control over the preceding 2–3 months. Also, in the linear mixed effect model for HbA1c, there may have been a slight time lag between the monthly adherence periods and the 8/12-week period reflected by HbA1c. This may have attenuated, rather than amplified, the observed differences. Furthermore, since adherence trajectories (CMA9) and outcome evaluated were constructed during the same follow-up period, residual temporal coupling could attenuate associations. Thus, some outcome measurements may precede later adherence changes. This timing overlap may attenuate estimates, so findings are interpreted as associations rather than causal effects. The same was the case of secondary outcome measures, LDL, blood pressure and BMI. These may be changed for reasons unrelated to OAD adherence (e.g., lifestyle, titration of statins/antihypertensives, care intensity). Additionally, clinical and demographic data were included only when essential fields were available. Records with missing clinical outcome measurements or incomplete patient information were excluded, and this may have introduced selection bias and could limit the generalizability of our findings. Lastly, the study cohort included only patients with a first recorded diagnosis of T2D, and we were unable to confirm whether they had been prescribed OADs prior to this diagnosis. Hence, a hypothetical prior exposure to an OAD treatment could have influenced adherence patterns that were not captured in this dataset.

## Conclusions

The results of our study provide valuable insights into the real-world dynamics of medication adherence among patients newly diagnosed type 2 diabetes (T2D) treated with oral antidiabetic drugs (OADs). This study identified four distinct adherence patterns in T2D patients and their diverse and significant association with main T2D clinical outcomes. Therefore, the findings confirm that high adherence in patients with T2D is associated with better clinical outcomes, particularly with significant reductions in HbA1c levels after one-year of treatment. This effect is even more evident in more complex patients with higher baseline values, suggesting that individuals with poorer initial control, likely receiving closer clinical monitoring, may achieve a perfect adherence behaviour. These results have practical implications for healthcare professionals, who should monitor not only whether patients initiate therapy but also how they maintain adherence over time (considering the three adherence phases). These four adherence trajectories (perfect, slow decline, low, slow increase) can support risk stratification and guide the intensity of follow-up, medication review, and education. Additionally, the previous identification of high-risk adherence patterns may be a useful tool for clinicians in targeting support strategies for patients. At the system level, integrating trajectory monitoring with routine risk-factor surveillance (HbA1c, LDL, blood pressure, BMI) could inform stratified care pathways and resource allocation. Hence, the use of adherence trajectories associated to clinical outcomes may inform the design of population-based monitoring systems and adherence-improving programmes. These findings support the integration of adherence profiling into chronic disease management frameworks and highlight the need for healthcare systems to include and consider data-driven approaches as a support for patient stratification, care coordination, and better resource allocation.

## Supporting information

**S1 Fig. Diagram highlighting study period and study design.** The figure illustrates the entire study period (1st January 2015–31 March 2021) including the recruitment period (1st January 2015–31 December 2019) and the individual one-year follow-up (FU) windows based on each patient's index date (first OAD prescription, represented by a black square) and ended maximum on 31 December 2020. Clinical parameters evaluation was registered for each patient within 3 months

before the index date (grey star) and within 3 months after the end of follow-up (grey hexagon). The adherence assessment within the follow-up window comprises the initiation (a), implementation (b), and discontinuation (c) phases. Each dashed line represents the sliding one-year follow-up window starting from the index date. The end of the study date was 31 March 2021. (DOCX)

**S2 Fig. Study flow chart.**
(DOCX)

**S3 Fig. Standardized cluster validity indices for selection of optimal number of adherence trajectories.** This figure illustrates the standardized values of these indices across different cluster solutions. The optimal number of adherence trajectories was determined by comparing model fit statistics across solutions ranging from two to six groups (k = 2–6). To ensure robustness and consistency, multiple criteria were applied, including: i) Calinski-Harabasz index (CH – criterion 1; CH2 – criterion 2; CH3 – criterion 3); ii) Ray-Turi index (criterion 4); iii) Davies-Bouldin index (criterion 5). Each of these metrics evaluates cluster validity from a different perspective, such as compactness (intra-cluster similarity) and separation (inter-cluster dissimilarity). The Calinski-Harabasz indices (CH2, CH3) and Ray-Turi index reach their highest or near-highest standardized scores at four clusters, indicating a favorable balance between within-group cohesion and between-group separation. While CH and Davies-Bouldin indices slightly favor k = 2 clusters, the overall multi-criteria profile supported k = 4 clusters as the preferred solution. Robustness and stability of these k = 4 clusters were further evaluated via bootstrap resampling, cluster-wise Jaccard similarities. Sensitivity analyses confirmed the selection [22–24]. (DOCX)

**S4 Fig. Longitudinal trajectories with a four adherence groups selection.** Each grey line represents the adherence trajectory. Each trajectory is the mean of monthly CMA9 over months 1–12 of a single T2D patient, grouped into one of the four adherence clusters (k = 4). Areas with denser concentration of lines indicate greater similarity in patient behavior, which guided the clustering algorithm. The solid black line shows the mean CMA trajectory for each group, while the grey shaded area denotes the standard deviation, capturing the variability around the mean. (DOCX)

**S5 Fig. Patients exceeding clinical outcomes T2D-related values stratified by adherence groups.**
(DOCX)

**S6 Fig. Changes in clinical outcomes T2D-related values from the beginning to the end of follow-up stratified by adherence groups.**
(DOCX)

**S7 Fig. Box-and-whisker plot correlating HbA1c levels at the start of follow-up and adherence groups.** Each box represents the interquartile range (IQR, 25th-75th percentile) of HbA1c values, with the horizontal line inside the box indicating the median HbA1c value. Whiskers extend to 1.5 × IQR, and outliers are shown as individual points. The dashed horizontal line represents the clinical HbA1c target threshold, set at 53 mmol/mol (7%). The plot highlights differences in HbA1c distribution across adherence groups, with group A (perfect adherence) showing higher median values at the baseline compared to the other adherence groups. (DOCX)

**S1 Table. Baseline patients' characteristics.**
(DOCX)

**S2 Table. Adherence phases and groups.**
(DOCX)

**S3 Table. Post-index insulin initiation within the one-year follow-up, by adherence trajectory group.** Insulin addition was defined as the first insulin prescription initiated after the index date (first OAD prescription) within the 12-month follow-up. Patients with any insulin before index were excluded from the overall T2D cohort. Percentages are within-group (the denominator is equal to the number of patients in each adherence group).Time at insulin addition is the number of days from index to the first post-index insulin prescription (mean value and standard deviation, SD).
(DOCX)

**S4 Table. Outcome Measures from the NHG Standaards for T2D monitoring.**
(DOCX)

**S5 Table. HbA1c (mmol/mol) levels by adherence trajectory: baseline vs end of one-year follow-up.**
(DOCX)

**S6 Table. Linear mixed-effects model for HbA1c by adherence trajectory and time.** The outcome HbA1c level measured in mmol/mol. Fixed effects: adherence group with Group A (perfect adherence) as reference, months since index date, and their interactions. Random intercepts and slopes at the patient level.
(DOCX)

**S7 Table. HbA1c slopes and association with adherence trajectories.** Slopes are derived from the linear mixed-effects model. Values represent the average monthly change in HbA1c (mmol/mol) over the one-year follow-up. The 12-month change is the monthly slope x 12.
(DOCX)

## Author contributions

**Conceptualization:** Liset van Dijk, Enrica Menditto.

**Data curation:** Sara Mucherino, Yvette Weesie.

**Formal analysis:** Sara Mucherino, Marcia Vervloet, Yvette Weesie, Enrica Menditto.

**Funding acquisition:** Valentina Orlando, Enrica Menditto.

**Investigation:** Sara Mucherino, Yvette Weesie, Liset van Dijk, Enrica Menditto.

**Methodology:** Sara Mucherino, Yvette Weesie, Enrica Menditto.

**Project administration:** Enrica Menditto.

**Software:** Sara Mucherino.

**Supervision:** Valentina Orlando, Marcia Vervloet, Liset van Dijk, Enrica Menditto.

**Validation:** Valentina Orlando, Marcia Vervloet, Liset van Dijk, Enrica Menditto.

**Visualization:** Valentina Orlando, Marcia Vervloet, Enrica Menditto.

**Writing – original draft:** Sara Mucherino.

**Writing – review & editing:** Valentina Orlando, Marcia Vervloet, Liset van Dijk, Enrica Menditto.

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
