## [Decision Letter · Decision Letter 0]

11 Sep 2025

Dear Dr. Mucherino,

Thank you for submitting your manuscript to PLOS ONE. After careful consideration, we feel that it has merit but does not fully meet PLOS ONE’s publication criteria as it currently stands. Therefore, we invite you to submit a revised version of the manuscript that addresses the points raised during the review process.

**We invited multiple reviewers and 3 have agreed with mixed reviews**

1 reject, 1 major revision and 1 minor revision

We look forward to receiving your revised manuscript.

Kind regards,

Yee Gary Ang, MBBS MPH

Academic Editor

PLOS ONE

Journal Requirements:

2. We note that you have indicated that there are restrictions to data sharing for this study. PLOS only allows data to be available upon request if there are legal or ethical restrictions on sharing data publicly.

For more information on unacceptable data access restrictions, please see http://journals.plos.org/plosone/s/data-availability#loc-unacceptable-data-access-restrictions.

3. In the online submission form you indicate that your data is not available for proprietary reasons and have provided a contact point for accessing this data. Please note that your current contact point is a co-author on this manuscript. According to our Data Policy, the contact point must not be an author on the manuscript and must be an institutional contact, ideally not an individual. Please revise your data statement to a non-author institutional point of contact, such as a data access or ethics committee, and send this to us via return email. Please also include contact information for the third party organization, and please include the full citation of where the data can be found.

Reviewers' comments:

Reviewer's Responses to Questions

**Comments to the Author**

1. Is the manuscript technically sound, and do the data support the conclusions?

Reviewer #1: Yes

Reviewer #2: Partly

Reviewer #3: Partly

2. Has the statistical analysis been performed appropriately and rigorously?

Reviewer #1: Yes

Reviewer #2: No

Reviewer #3: No

3. Have the authors made all data underlying the findings in their manuscript fully available?

Reviewer #1: Yes

Reviewer #2: No

Reviewer #3: Yes

4. Is the manuscript presented in an intelligible fashion and written in standard English?

Reviewer #1: Yes

Reviewer #2: No

Reviewer #3: Yes

Reviewer #1: The data and the topic of the manuscript is interesting. However, the manuscript itself is difficult to read because of long paragraphs and complex sentences. The authors should condense sentences throughout the text. Abstract should be clearly structured.

The definition of follow-up is not clearly explained in methods section. Figure 1 clarifies this, but some of the information of figure 1 should be explained also in the text.

Does baseline characteristics refer to index date (table 1)?

Because the adherence is defined based on the prescription of medication, the system of prescription data should be clarified. For how long period is one prescription? Is there any system to check whether patient has purchased the medication prescribed?

It is not clear for me, why patient with only one OAD prescription had to be excluded. Even this is explained, I don’t understand the argument. (lines 106-108).

Reviewer #2: Although the research question addressed in the manuscript is important, the reporting of the approach is confusing, and there are multiple inconsistencies between the methods and results. My detailed concerns are outlined below:

1) Study Design

The design of the study is not clearly described.

Baseline definition: The definition of “baseline” is incorrect.

Lines 120–122: Profiles of subjects within each adherence trajectory were examined through an analysis of their baseline characteristics, including gender and age, their drug utilization patterns, and the addition of insulin therapy.

With the index defined as initiation of OAD, what is meant by “addition of insulin therapy”? If patients were on insulin before OAD initiation, they should have been excluded. If insulin was added after OAD initiation, then this addition occurred outside of the baseline period.

After carefully reading the manuscript, tables, and the responses to reviewer comments, it seems the design is actually cross-sectional for assessing the association between adherence and outcomes. This should be clearly stated.

2) Adherence Measures

The definitions and implementation of adherence measures are confusing. A few examples:

Lines 160–161: “Prescription histories for each single OAD medication over an observation period of 365 days (1 year) were calculated.”

Line 202: “In the persistence phase, 84.6% of the overall T2D cohort remained on therapy at the end of the one-year follow-up.”

It is unclear whether prescription history was captured across the full one-year period or only within the persistence phase.

In response to Reviewer 2, Q7, you stated:

“The group-based trajectory approach was applied to daily adherence data, specifically the continuous multiple-interval measure of medication availability (CMA), over a 12-month follow-up period…”

However, the corresponding section in lines 112–188 does not reflect this explanation. I did not realize adherence was measured monthly until I reached this response. The manuscript itself should make this point clear.

It is also unclear whether persistence and adherence were measured at the patient level or the drug level. Patient-level measurement is appropriate; however, the current description suggests drug-level measurement. If this is the case, how was switching handled? Were patients switching from one OAD to another classified as nonadherent?

3) Linear Mixed-Effects Regression Model

The description of the mixed-effects model is insufficient.

The methods section does not specify whether baseline characteristics (e.g., disease severity) were controlled.

The results section lacks a table summarizing the regression results. The figure provided is not sufficient to determine whether the observed outcomes truly differ across groups once baseline covariates are appropriately adjusted.

Conclusion

Overall, while the manuscript addresses a significant and relevant research question, substantial revisions are needed to clarify the study design, strengthen the description and interpretation of adherence measures, and provide a more transparent and detailed account of the statistical analysis. These changes are essential for the work to be interpretable and credible.

Reviewer #3: The retrospective design using prescription refill data does not allow inference on whether medication was actually consumed. This fundamental limitation undermines the central claims linking adherence trajectories with improved clinical outcomes.

The definition of initiation as “at least two prescriptions” is arbitrary and excludes patients who discontinued legitimately after one prescription. This exclusion introduces bias and potentially misclassifies patients, weakening external validity.

The justification for selecting four adherence clusters is insufficient. Although indices (Calinski-Harabasz, Davies-Bouldin) were applied, the process appears exploratory and lacks robustness testing (e.g., bootstrapping, validation in external datasets). Without external replication, the stability of clusters and generalizability of identified trajectories are questionable.

Key confounders such as socioeconomic status, educational attainment, and health literacy—well-established determinants of adherence—are absent. Relying solely on comorbidity indices (Charlson, multimorbidity count) does not sufficiently control for patient-related factors, leading to potentially spurious associations between “perfect adherence” and better outcomes.

Clinical outcomes were restricted to HbA1c, LDL, blood pressure, and BMI. Other relevant T2D outcomes such as microvascular complications, hospitalizations, and hypoglycemia were ignored, narrowing the clinical relevance of the findings.

HbA1c reflects glycaemic control over 2–3 months, yet persistence and implementation measures were calculated annually. This temporal discordance weakens causal interpretations.

The authors claim that trajectory modelling is novel, yet several recent studies have applied similar approaches in T2D and other chronic diseases. The manuscript does not convincingly demonstrate methodological innovation or clinical impact beyond confirming established knowledge.

**Do you want your identity to be public for this peer review?** For information about this choice, including consent withdrawal, please see our Privacy Policy

Reviewer #1: No

Reviewer #2: No

Reviewer #3: No

---

## [Author Response · Author response to Decision Letter 1]

6 Oct 2025

Response to Reviewers

PONE-D-25-41866

Medication adherence trajectories to predict risk factors and clinical outcomes in type 2 diabetes treatment

Academic Editor Comments

Dear Dr. Mucherino, thank you for submitting your manuscript to PLOS ONE. After careful consideration, we feel that it has merit but does not fully meet PLOS ONE’s publication criteria as it currently stands. Therefore, we invite you to submit a revised version of the manuscript that addresses the points raised during the review process.

Response: Dear Academic Editor, we appreciate your consideration and the valuable input from the three reviewers. We have addressed all reviewer comments and prepared a substantially revised version of the manuscript and supplementary material. Below in this report you’ll find all the detailed responses to reviewers also addressed in the manuscript.

Journal Requirements

Comment 1. Please ensure that your manuscript meets PLOS ONE's style requirements, including those for file naming. The PLOS ONE style templates can be found at

Response: We have revised the submission to comply with PLOS ONE style and file-naming requirements.

Comment 2. We note that you have indicated that there are restrictions to data sharing for this study. PLOS only allows data to be available upon request if there are legal or ethical restrictions on sharing data publicly.

For more information on unacceptable data access restrictions, please see http://journals.plos.org/plosone/s/data-availability#loc-unacceptable-data-access-restrictions.

Response: Data of the Nivel Primary Care Database (Nivel-PCD) contain potentially identifying or sensitive patient information. Access to data in Nivel-PCD is subject to Nivel-PCD governance codes. Requests for access to the data can be directed at datarequest@nivel.nl. Restrictions include approval by the appropriate Nivel-PCD governance bodies (privacy committee and steering committee).

Comment 3. In the online submission form you indicate that your data is not available for proprietary reasons and have provided a contact point for accessing this data. Please note that your current contact point is a co-author on this manuscript. According to our Data Policy, the contact point must not be an author on the manuscript and must be an institutional contact, ideally not an individual. Please revise your data statement to a non-author institutional point of contact, such as a data access or ethics committee, and send this to us via return email. Please also include contact information for the third party organization, and please include the full citation of where the data can be found.

Response: Data of the Nivel Primary Care Database (Nivel-PCD) contain potentially identifying or sensitive patient information. Access to data in Nivel-PCD is subject to Nivel-PCD governance codes. Requests for access to the data can be directed at datarequest@nivel.nl. Restrictions include approval by the appropriate Nivel-PCD governance bodies (privacy committee and steering committee).

Comment 4. If the reviewer comments include a recommendation to cite specific previously published works, please review and evaluate these publications to determine whether they are relevant and should be cited. There is no requirement to cite these works unless the editor has indicated otherwise.

Response: Thanks for the notice. No citations were recommended by the reviewers.

Reviewers’ Comments

Reviewer #1 Comments

Comment 1. The data and the topic of the manuscript is interesting. However, the manuscript itself is difficult to read because of long paragraphs and complex sentences. The authors should condense sentences throughout the text. Abstract should be clearly structured.

Response: Dear Reviewer, many thanks for the valuable comments provided. We understand this observation. We have revised the whole manuscript for clarity, shortening long sentences, splitting dense paragraphs, and removing redundant phrasing. We also reformatted the Abstract into a fully structured format (Background, Methods, Results, Conclusions).

Comment 2. The definition of follow-up is not clearly explained in methods section. Figure 1 clarifies this, but some of the information of figure 1 should be explained also in the text.

Response: We have now made the follow-up definition explicit also in the methods section. We now aligned the text with the scheme shown in Figure 1.

Comment 3. Does baseline characteristics refer to index date (table 1)?

Response: Yes. Baseline characteristics are assessed at the index date, defined as the date of the first OAD prescription. We better specified this in the new revised version of the manuscript.

Comment 4. Because the adherence is defined based on the prescription of medication, the system of prescription data should be clarified. For how long period is one prescription? Is there any system to check whether patient has purchased the medication prescribed?

Response: We fully understand this important clarification request. In this study (based on Nivel-PCD database) we observed general-practice prescribing records (prescription date, drug/ATC code, quantity/instructions when available, diagnosis, and so on). We do not have pharmacy dispensing claims. This is of course a study limitation that should be considered, primarily related to potential biases, related, of course, to the nature of prescription data itself. To address this, we considered at least two prescriptions within the observation period as a proxy for treatment initiation and continuation but, we cannot directly verify whether a patient really collected or ingested the medication. This aspect is highlighted in the limits’ declaration in the discussion section.

Moreover, the duration of all OAD prescriptions is not fixed, as each OAD have a different coverage period. Hence, to standardize exposure, we derived days’ supply from the recommended daily dosing for each OAD class using national sources already cited in the manuscript (NHG guideline and the Medicines Information Bank of the Dutch MEB). For example, sulfonylureas, thiazolidinediones, DPP-4 inhibitors, and SGLT2 inhibitors are typically once daily; combinations/α-glucosidase inhibitors 1-3 times/day; biguanides 2-3 times/day (extended-release once daily). We used these dosing rules to compute expected coverage per prescription and to construct monthly CMA9 trajectories. This is reported in the manuscript within the Statistical Analyses section.

Additionally, because dispensing is unavailable, persistence phase was computerised from prescribing patterns. A patient was classified non-persistent when no subsequent prescription was observed or when the interval to the next prescription exceeded 2x the expected coverage of the previous one. Patients with only one OAD prescription during follow-up were reported under initiation but excluded from persistence estimation to avoid misclassification (eg, primary non-initiation versus very long coverage cannot be distinguished with a single prescription).We’ve now better detailed and described this process within the section “Trajectory modelling and medication adherence estimation” (totally revised).

Comment 5. It is not clear for me, why patient with only one OAD prescription had to be excluded. Even this is explained, I don’t understand the argument. (lines 106-108).

Response: We understand the question and probably we explained the process of treating patients with a single prescription in an unclear manner, in the first version of the manuscript. To be clearer, we did not count patients with only 1 OAD prescription just in the persistence and trajectory analyses. This is because persistence is a time-to-event construct that requires at least two prescriptions to observe continued exposure and to define whether/when a permissible gap is exceeded. With a single prescription we cannot distinguish among: primary non-initiation, immediate discontinuation after the first script, very long coverage, or unobserved switching outside the index OAD. All these factors would misclassify persistence if we imposed an assumption. The same issue affects implementation trajectories (CMA9 in monthly windows). Indeed, A single prescription provides insufficient longitudinal information to form a stable trajectory and would influence the grouping. But it’s important to clarify that, these patients were not totally excluded from the study. These patients were counted under initiation (reported in the Results and SI) and described at baseline but not counted only from persistence and trajectory modelling to avoid differential misclassification. They represent a small fraction of the cohort (0.9%), so this decision does not materially affect the main estimates.

We understand that this justification was not well argued in the previous version of the manuscript, so we have detailed these specifications in the current revision of the manuscript.

Reviewer #2 Comments

Comment 1: Although the research question addressed in the manuscript is important, the reporting of the approach is confusing, and there are multiple inconsistencies between the methods and results. My detailed concerns are outlined below

Response: Dear Reviewer, thank you for this careful assessment. We appreciate the detailed comments provided in your review. Hence, considering your feedback, we have carried out an overall revision of the manuscript. Mainly, we clarified the analytical approach used step-by-step (data source, cohort construction, adherence definitions across initiation/implementation/discontinuation, CMA9 computation in monthly windows, trajectory modelling and selection criteria, and regression modelling). We have also aligned the methods and results so that each estimate reported now corresponds in a clear way. We standardized all the terminology (such as the index date, the study period and follow-up etc). Tables have been revised for readability, as per tables and figures’ footnotes (we included clearer descriptions). We also expanded the Supplementary Information with additional methodological detail (trajectory diagnostics and selection rationale, definitions of covariates, handling of missing data, and the exact clinical-window alignment for pre/post outcomes). Finally, we performed a full language edit to shorten long sentences. We hope these changes directly address your concerns about confusing reporting and any inconsistencies between methods and results. We are thankful for the improvements that your comments provided.

Comment 2: 1) Study Design. The design of the study is not clearly described. Baseline definition: The definition of “baseline” is incorrect. Lines 120–122: Profiles of subjects within each adherence trajectory were examined through an analysis of their baseline characteristics, including gender and age, their drug utilization patterns, and the addition of insulin therapy. With the index defined as initiation of OAD, what is meant by “addition of insulin therapy”? If patients were on insulin before OAD initiation, they should have been excluded. If insulin was added after OAD initiation, then this addition occurred outside of the baseline period.

Response: Thank you for pointing this out. We agree and recognize this mistake. Indeed, in the first submission we added the number of patients starting insulin and time at insulin addition start improperly in Table 1. While, Table 1 (and also Supplementary Table S1) have to report baseline characteristics only of the overall OAD cohort also stratified by adherence groups.

We now amended these corrections. Specifically, in the revised manuscript we have restricted Table 1 to true baseline variables (sex, age, comorbidity, concomitant medicines and diagnosis, etc), and removed the insulin-start variable. Additionally, we now clarified in the manuscript that baseline values refer to the assessment window within 90 days before the index date (first OAD prescription). Also, we added the clarification on how we evaluated insulin addition. Namely, it was identified as any insulin prescription initiated after the index date (date of first OAD prescription) within the 12-month follow-up. We also make explicit in methods that the overall T2D cohort excluded patients with insulin before OAD initiation. Hence, insulin prescription was evaluated only after index as a follow-up characteristic and included in the regression as a post-index indicator, not as a baseline attribute. We also added a separate table to clarify this in the Supplementary material (new Table S4). You can find all the corresponding clarifications, additions and modifications in this new version of the manuscript and supplementary file.

Comment 3: After carefully reading the manuscript, tables, and the responses to reviewer comments, it seems the design is actually cross-sectional for assessing the association between adherence and outcomes. This should be clearly stated.

Response: We understand why parts of the first version could give that impression. However, the study is a retrospective longitudinal cohort, not cross-sectional. In this study we selected patients at the index date (date of the first OAD prescription) and followed each patient for 12 months. Hence, adherence was modelled longitudinally, using monthly CMA9 windows over the full follow-up to build the trajectory groups (implementation). Persistence/discontinuation is derived from the same time-stamped prescription series with a permissible-gap rule. Also, baseline values were assessed in the 90 days before index date, and values at the end of follow-up were collected in the 90 days after month 12. Although this, we recognise that descriptive summaries of values at the end of follow-up by trajectory may read as “snapshots” if timing is not explicit. We have therefore revised the manuscript to make the longitudinal design and time windows clearer. We also have now clarified that these values are descriptive longitudinal summaries.

We additionally now recognised that, as adherence trajectories and outcomes are constructed over the same 12-month horizon, we avoid causal language and interpret differences as associations. We corrected the whole manuscript as appropriate. All these clarifications now appear in the methods and in the limitations of the revised manuscript.

Comment 4: 2) Adherence Measures. The definitions and implementation of adherence measures are confusing. A few examples:

Lines 160–161: “Prescription histories for each single OAD medication over an observation period of 365 days (1 year) were calculated.”

Line 202: “In the persistence phase, 84.6% of the overall T2D cohort remained on therapy at the end of the one-year follow-up.”

It is unclear whether prescription history was captured across the full one-year period or only within the persistence phase.

Response: In the revised manuscript we now clarified t

---

## [Decision Letter · Decision Letter 1]

27 Oct 2025

Dear Dr. Mucherino,

Thank you for submitting your manuscript to PLOS ONE. After careful consideration, we feel that it has merit but does not fully meet PLOS ONE’s publication criteria as it currently stands. Therefore, we invite you to submit a revised version of the manuscript that addresses the points raised during the review process.

We look forward to receiving your revised manuscript.

Kind regards,

Yee Gary Ang, MBBS MPH

Academic Editor

PLOS ONE

Journal Requirements:

Additional Editor Comments:

Please read the reviewers carefully and resubmit if you are able to address them

Reviewers' comments:

Reviewer's Responses to Questions

**Comments to the Author**

Reviewer #1: (No Response)

Reviewer #2: All comments have been addressed

2. Is the manuscript technically sound, and do the data support the conclusions?

Reviewer #1: Partly

Reviewer #2: Yes

3. Has the statistical analysis been performed appropriately and rigorously?

Reviewer #1: Yes

Reviewer #2: Yes

4. Have the authors made all data underlying the findings in their manuscript fully available?

Reviewer #1: Yes

Reviewer #2: Yes

5. Is the manuscript presented in an intelligible fashion and written in standard English?

Reviewer #1: (No Response)

Reviewer #2: No

Reviewer #1: The text of Abstract has improved and now clearly structured.

The definition of follow-up is now clarified in methods section

The system of prescription differs in different countries. Therefore, the prescription system in Netherlands should be clarified in methods section. For how long period is one prescription? For example, in my country prescription of insulin is for one year and for oral medication it is two years. After that the physicians have to renew the prescription of each medication. Please clarify the detailed prescription system in Netherlands.

The authors have edited the text, but some of the paragraphs are still complex. The authors could further condense sentences to make it easier to read.

Reviewer #2: (No Response)

**Do you want your identity to be public for this peer review?** For information about this choice, including consent withdrawal, please see our Privacy Policy

Reviewer #1: No

Reviewer #2: No

---

## [Author Response · Author response to Decision Letter 2]

31 Oct 2025

Reviewers’ Comments

Reviewer #1 Comments

Comment 1. The text of Abstract has improved and now clearly structured. The definition of follow-up is now clarified in methods section.

Response: Dear Reviewer, many thanks for your positive feedback. In this revised submission, we have also addressed the additional points raised in your second round of comments, with corresponding changes integrated.

Comment 2. The system of prescription differs in different countries. Therefore, the prescription system in Netherlands should be clarified in methods section. For how long period is one prescription? For example, in my country prescription of insulin is for one year and for oral medication it is two years. After that the physicians have to renew the prescription of each medication. Please clarify the detailed prescription system in Netherlands.

Response: Dear Reviewer, we understand this specification. We added this concept in the method section. Specifically, in Dutch primary care, chronic therapies are issued as repeat prescriptions (“herhaalrecept”). In standard practice, a repeat authorization typically covers up to 12 months. Community pharmacies, however, dispense the medications approximately 1-3 months per prescription, based on the insurer's policy and clinical opinion. Insulins and oral antidiabetic drugs follow the same framework. The prescription authorisation may cover a year, but the dispensed quantity per issue is generally limited to ~3 months. These explain why prescription dates recur periodically in the electronic record and support our construction of coverage periods from successive prescription records. We therefore computed days of supply from the prescribed regimen and dates, allowing carry-over, and assessed persistence via a permissible-gap rule over the 12-month follow-up. We have now specified this in the methods clarifying these points.

Comment 3. The authors have edited the text, but some of the paragraphs are still complex. The authors could further condense sentences to make it easier to read.

Response: Dear Reviewer, thanks for this remark. We conducted an additional full read-through of the main text and further simplified several long or complex sentences to improve readability.

---

## [Decision Letter · Decision Letter 2]

15 Nov 2025

Dear Dr. Mucherino,

Thank you for submitting your manuscript to PLOS ONE. After careful consideration, we feel that it has merit but does not fully meet PLOS ONE’s publication criteria as it currently stands. Therefore, we invite you to submit a revised version of the manuscript that addresses the points raised during the review process.

We look forward to receiving your revised manuscript.

Kind regards,

Yee Gary Ang, MBBS MPH

Academic Editor

PLOS ONE

Journal Requirements:

Additional Editor Comments:

We only managed to get 1 reviewer who recommended major revision

Please assess and resubmit if you are able to address the issues raised.

Reviewer's Responses to Questions

**Comments to the Author**

Reviewer #4: (No Response)

2. Is the manuscript technically sound, and do the data support the conclusions?

Reviewer #4: Partly

3. Has the statistical analysis been performed appropriately and rigorously?

Reviewer #4: I Don't Know

4. Have the authors made all data underlying the findings in their manuscript fully available?

Reviewer #4: No

5. Is the manuscript presented in an intelligible fashion and written in standard English?

Reviewer #4: Yes

Reviewer #4: General Comments

This is my first review of this paper assessing medication adherence trajectories and their association with characteristics and clinical outcomes in type 2 diabetes. The writing is generally clear, and the objectives and methods are reasonably sound. I have several major and minor comments to improve clarity and relevance of the manuscript.

Specific comments

Here are improvements for each point:

1. Title: The current title “Medication adherence trajectories to predict risk factors and clinical outcomes in type 2 diabetes treatment.” is misleading because the trajectories are assessed concurrently with clinical outcomes. Therefore, the authors cannot claim that adherence trajectories predict outcomes; they can only state that associations exist. Additionally, the manuscript uses the words correlation and association interchangeably. The authors should be more precise about the type of analysis performed, the specific resulting measures and they should interpret them correctly.

2. The previous reviewer asked clarifications on how the prescription system works in the Netherlands, particularly regarding prescription duration. The authors explained that “therapies are issued as repeat prescriptions that typically covers up to 12 months and that community pharmacies, dispense these medications approximately 1-3 months per prescription ». However, it remains unclear what type of data are available in the Nivel Primary Care Database (Nivel-PCD): prescription data (what the doctor prescribed), dispensation data (what the pharmacist provided) or both? The authors should use precise terminology (e.g., medication supply, medication claim, medication dispensed, medication prescribed) throughout the manuscript to avoid confusion.

3. Line 79: Are you certain that 90% of 1.1 million patients in 2019 had type 2 diabetes? Or do you mean that among patients with diabetes, 90% had type 2? This needs clarification.

4. Line 81-84: It is unclear whether you are referring to the Nivel Primary Care Database or your study population when discussing inclusion criteria.

5. Line 107-108: Why was the initiation phase defined as at least two prescriptions within one year? Initiation typically refers to when the patient takes the first dose of the prescribed medication. To assess initiation accurately, both prescription and dispensation data are needed. Otherwise, using the term “initiation” is misleading. This should be clarified.

6. Lines 116-117: The authors state “Patients with only one prescription during follow-up were reported under initiation but excluded from persistence and implementation estimations.” While it is understandable why implementation was not calculated for these patients, why was persistence not assessed? These patients clearly did not persist with their treatment beyond the initial dispensation period.

7. Lines 124-126: The cluster analysis is poorly explained and should be clarified.

8. Lines 162-163: This limitation - that adherence and outcomes were measured over the same period and that outcomes may have occurred before exposure - should be explicitly stated in the limitations section.

9. Line 178: Not all sulfonylureas are taken once daily. This should be noted as a limitation, given that the database lacks information on directions for use or medication days’ supply.

10. Lines 191-194: This section is difficult to understand, even for readers familiar with this methodology. It should be rewritten to improve clarity.

11. Line 210: State that p-values<0.05 were “considered” statistically significant.

12. Lines 227-228: Again, this does not represent initiation per se. This should be revised.

13. Line 233: FU (means follow-up)? This term should be written in full for clarity.

14. Lines 244-245: This sentence contains two instances of “after”. One should be removed.

15. Lines 281-290: The authors report LDL, BMI and blood pressure control at the beginning and the end of the 12-month follow-up period. This seems questionable. What is the rationale for linking adherence to OAD with LDL, BMI and blood pressure control? The authors claim they are the only ones to have perform that type of analysis, which they say is a strength of their work (lines 373-377). But what is the novelty/strength if it does not make sense? This is a major flaw of this paper, certainly not a strength.

16. Lines 295-305: Results in this section are reported with three decimal places, whereas the rest of the manuscript uses two. This should be standardized.

17. Lines 313-318: These sentences and their connections are unclear and should be revised for better flow.

18. Line 421-423: The authors state that their “findings confirm that high adherence in patients with type 2 diabetes is associated with better clinical outcomes…”. Was a study needed to confirm this? What new insights does this study provide and how can they be applied?

19. References 41 and 46 are duplicates and should be corrected.

**Do you want your identity to be public for this peer review?** For information about this choice, including consent withdrawal, please see our Privacy Policy

Reviewer #4: **Yes:** Line Guénette

---

## [Author Response · Author response to Decision Letter 3]

29 Nov 2025

Reviewer #4 Comments

General comment: This is my first review of this paper assessing medication adherence trajectories and their association with characteristics and clinical outcomes in type 2 diabetes. The writing is generally clear, and the objectives and methods are reasonably sound. I have several major and minor comments to improve clarity and relevance of the manuscript.

Response: Dear Reviewer, we greatly appreciate both your general and specific comments on the manuscript. We have made the requested changes to improve the clarity of the manuscript. We hope you will appreciate the revised version of the resubmitted manuscript. Below you will find point-by-point responses to each of your comments. Thank you.

Comment 1. Title: The current title “Medication adherence trajectories to predict risk factors and clinical outcomes in type 2 diabetes treatment.” is misleading because the trajectories are assessed concurrently with clinical outcomes. Therefore, the authors cannot claim that adherence trajectories predict outcomes; they can only state that associations exist. Additionally, the manuscript uses the words correlation and association interchangeably. The authors should be more precise about the type of analysis performed, the specific resulting measures and they should interpret them correctly.

Response: Dear Reviewer, thanks to note this important point. We agree that our study assesses concurrent associations rather than prediction or causal effects. We have therefore: i) revised the title to remove “predict” and ii) standardised wording across the manuscript to use “association/associated with” and avoid “predict/impact”. Also we now used “association” rather than “correlation” (except when referring to correlation as a statistical quantity). The manuscript now explicitly states the observational nature of the design and that we refrain from causal claims. These changes are now reflected in the title and within the whole manuscript.

Comment 2. The previous reviewer asked clarifications on how the prescription system works in the Netherlands, particularly regarding prescription duration. The authors explained that “therapies are issued as repeat prescriptions that typically covers up to 12 months and that community pharmacies, dispense these medications approximately 1-3 months per prescription ». However, it remains unclear what type of data are available in the Nivel Primary Care Database (Nivel-PCD): prescription data (what the doctor prescribed), dispensation data (what the pharmacist provided) or both? The authors should use precise terminology (e.g., medication supply, medication claim, medication dispensed, medication prescribed) throughout the manuscript to avoid confusion.

Response: Dear Reviewer, we agree that a coherent terminology is essential for the clarity of the manuscript. Please, consider that the Nivel Primary Care Database (Nivel-PCD) contains general practice electronic health record data, including prescribed medications recorded by GPs (date, drug/ATC, dose instructions, and quantity/duration). Hence, Nivel-PCD does not contain pharmacy dispensing claims. We have now standardised wording throughout the manuscript to be sure to avoid terms implying dispensing (such us medication refill etc). Moreover, days of supply and coverage were derived from the GP prescription record (prescribed daily dose and recorded duration/quantity), and adherence was computed on that basis. Persistence was assessed on the same prescription series using a permissible-gap rule, and switching to a different OAD with cessation of index prescriptions was treated as discontinuation of the index OAD. We have also now clarified how repeat prescriptions work in Dutch primary care. Generally, GPs authorise repeat therapy (typically for up to 12 months), while community pharmacies dispense in shorter amounts (often 1–3 months per issue). Finally, the database we used for the study captured the prescription authorisations and dates from GPs, not the pharmacy dispenses. All of these clarifications have been now added to the method section.

Comment 3. Line 79: Are you certain that 90% of 1.1 million patients in 2019 had type 2 diabetes? Or do you mean that among patients with diabetes, 90% had type 2? This needs clarification.

Response: We understand that the sentence may have caused some confusion. Referring to reference no. 19 of CBS statistics Netherlands, we confirm that the percentage referred to the total number of diabetic patients. We have therefore reworded the sentence more precisely as follows: “In the Netherlands, type 2 diabetes accounts for 82% of all diabetes cases [19], with negative consequences for public health [20–22]”.

Comment 4. Line 81-84: It is unclear whether you are referring to the Nivel Primary Care Database or your study population when discussing inclusion criteria.

Response: Yes definitely. We referred to our data source, which is Nivel-PCD. We have now specified the data source more clearly in the inclusion criteria, where suggested. Thanks.

Comment 5. Line 107-108: Why was the initiation phase defined as at least two prescriptions within one year? Initiation typically refers to when the patient takes the first dose of the prescribed medication. To assess initiation accurately, both prescription and dispensation data are needed. Otherwise, using the term “initiation” is misleading. This should be clarified.

Response: We understand the reason you are raising this point. According the EMERGE guidelines, initiation is the patient’s first dose and that confirming it would require dispensing (or intake) information. Albeit this, our dataset contains prescription records issued by GPs, not pharmacy dispensing claims. That said, as no dispensing data was available, we used a proxy: having a second prescription from GPs presumably means that the first dose was dispensed and taken. We now better declared in the study methods that “Initiation phase was calculated as the occurrence of at least two prescriptions of the index oral antidiabetic drug (OAD) within the one-year follow-up period, since dispensing data were not available” to avoid a misleading use of the term. We have also now better specified within the study limitation section that “The analyses were based on a data source which contains prescription records of the whole Dutch population. However, prescription records cannot confirm whether patients took or not their prescribed medication. To address this, we considered at least two prescriptions within the observation period as a proxy for treatment initiation and continuation. Hence, days of supply were built on OADs-prescriptions.”.

Comment 6. Lines 116-117: The authors state “Patients with only one prescription during follow-up were reported under initiation but excluded from persistence and implementation estimations.” While it is understandable why implementation was not calculated for these patients, why was persistence not assessed? These patients clearly did not persist with their treatment beyond the initial dispensation period.

Response: As according to the EMERGE guidelines, persistence is a time-to-event, hence, in this study, we measured it based on a permissible-gap rule, which requires at least two prescriptions to identify when a gap first exceeds 2× the expected coverage. With only one prescription, we cannot observe a discontinuation date without imputing it from a single record (which risks misclassification given heterogeneity in days-supply and possible unobserved switches). For this reason, patients with one prescription were reported under treatment start evaluation (based on prescribing data) but excluded from persistence and trajectory estimation. They represent about 0.9% of the cohort, so the impact on overall estimates is minimal. We have now made this rationale more explicit in methods.

Comment 7. Lines 124-126: The cluster analysis is poorly explained and should be clarified.

Response: Dear reviewer, we’ve now more in depth explained the cluster analysis approach where indicated. More details are also described within the statistical analysis. We added these paragraph as follows: “Clustering method used consisted of a set of multivariate techniques aimed at recognising and grouping homogeneous items in a data set by using non-parametric statistics [10, 24–26]. The clustering approach grouped patients with similar month-to-month adherence rates over the first year of the OAD-therapy. Each cluster was summarized by a centroid, i.e., the typical adherence curve formed by averaging the profiles of its members. Then, the clusters were labelled by the shape of their centroid to reflect recognizable behaviours in adherence. Hence, cluster analysis for adherence implementation measurement allowed the identification of groups of T2D patients with common characteristics (e.g., potential determinants or predictors of risk) [14, 24–26]. This approach linked heterogeneous individual adherence patterns into a small set of clinically interpretable behaviours.”.

Comment 8. Lines 162-163: This limitation - that adherence and outcomes were measured over the same period and that outcomes may have occurred before exposure - should be explicitly stated in the limitations section.

Response: Yes, we understand that there is the possibility that some outcome measurements may occur before subsequent adherence changes. We’ve now more explicitly stated this in the limitation section.

Comment 9. Line 178: Not all sulfonylureas are taken once daily. This should be noted as a limitation, given that the database lacks information on directions for use or medication days’ supply.

Response: Thank you for this helpful point. We agree that dosing within classes (such as sulfonylureas) varies by molecule and formulation. We have to specify that in our dataset, days of supply were primarily derived from prescription-level fields: namely the recorded prescription duration (days) and the prescribed daily dose. We of course checked that the posology was in accordance with the national guidelines. Also, when these fields were missing, we infer coverage using molecule/formulation-specific dosing from national guidance and product information, together with the timing of successive prescriptions. For examples, for sulfonylureas, modified-release products were treated as once-daily, whereas immediate-release formulations were treated as once- to twice-daily. This was consistent with product characteristics. We understand that this approach must be better specified within the manuscript, hence, we have now revised the methods section to reflect this and also added limitation noting potential of coverage.

Comment 10. Lines 191-194: This section is difficult to understand, even for readers familiar with this methodology. It should be rewritten to improve clarity.

Response: We’ve totally rewritten the paragraph as follows: “Cluster stability was assessed in three steps. First, k-means was run with 100 random initialisations to reduce dependence on starting values. Second, bootstrap resampling was applied (200 samples with replacement). For each bootstrap sample, the solution was compared with the full-sample solution using the Jaccard index for cluster membership. Specifically, values ≥0.75 were considered stable and values 0.60–0.75 moderately stable. Third, a split-sample check was performed: centroids were estimated in a random 50% training subset, and the remaining 50% were assigned to the nearest centroid. Finally, agreement with the full-sample assignment was then evaluated.”

Comment 11. Line 210: State that p-values<0.05 were “considered” statistically significant.

Response: We added this clarification. Thanks.

Comment 12. Lines 227-228: Again, this does not represent initiation per se. This should be revised.

Response: We addressed this point in response to the earlier comment on the definition of initiation. We have now clarified in the methods that initiation in this study refers to treatment start based on prescribing data, not confirmed medication intake, and that this definition aligns with what can be observed in prescription records. We also acknowledged this better within the limitation section.

Comment 13. Line 233: FU (means follow-up)? This term should be written in full for clarity.

Response: Yes, we have now corrected it and written it out in full.

Comment 14. Lines 244-245: This sentence contains two instances of “after”. One should be removed.

Response: We’ve amended it. Thanks.

Comment 15. Lines 281-290: The authors report LDL, BMI and blood pressure control at the beginning and the end of the 12-month follow-up period. This seems questionable. What is the rationale for linking adherence to OAD with LDL, BMI and blood pressure control? The authors claim they are the only ones to have perform that type of analysis, which they say is a strength of their work (lines 373-377). But what is the novelty/strength if it does not make sense? This is a major flaw of this paper, certainly not a strength.

Response: We agree that HbA1c is the primary glycaemic outcome for OAD adherence, and we do not claim that OAD adherence causes changes in LDL, blood pressure or BMI. The aim of this study was different: in routine T2D care, cardiometabolic risk-factor control (HbA1c, LDL, BP, weight) is monitored together and managed multifactorially. Many patients in our study cohort received concomitant lipid-lowering and antihypertensive therapies, and adherence to OADs often coexists with broader treatment adherence and care engagement. Also for this reason, we treated LDL, BP and BMI as secondary, descriptive outcomes, used to profile clinical risk-factor control across adherence groups (not to infer drug-specific effect).

To avoid any implication of causality, we have now revised the manuscript in tha way: i) position LDL, BP and BMI explicitly as exploratory secondary outcomes. ii) Make clear that analyses are associational and descriptive. iii) Acknowledge that changes in these outcomes may reflect multiple concurrent influences (lifestyle, intensification of non-glycaemic medications, clinical follow-up), beyond OAD adherence.

All these addition/modifications were made within the introduction, methods, results and limitation section.

Comment 16. Lines 295-305: Results in this section are reported with three decimal places, whereas the rest of the manuscript uses two. This should be standardized.

Response: We’ve now amended it. We changed everything to two decimal places to be consistent throughout the manuscript. Thanks.

Comment 17. Lines 313-318: These sentences and their connections are unclear and should be revised for better flow.

Response: Dear reviewer, we understand this clarification request. We’ve now changed the paragraph as follows: “This study mapped medication adherence across its three phases (initiation, implementation, and persistence/discontinuation) during the first year of treatment in newly diagnosed T2D receiving OADs. The three adherence phases were evaluated in accordance with the most recent adherence guidelines (the EMERGE guidelines [5] and the ABC taxonomy [6]). Using longitudinal trajectories, four common adherence behaviours were identified (perfect adherence, slow decline in adherence, low adherence, slow increase in adherence). Higher-adherence patterns were associated with better risk-factor control over time, particularly lower HbA1c and LDL cholesterol. The OAD-therapy initiation was high (99%), and one-year persistence was consistent with prior T2D reports (70–80%) when considered across adherence subgroups, with marked variation between behaviours [32–34].”

Comment 18. Line 421-423: The authors state that their “findings confirm that high adherence in patients with type 2 diabetes is associated with better clinical outcomes…”. Was a study needed to confirm this? What new insights does this study provide and how can they be applied?

Response: We know that the association between higher adherence and better glycaemic control is well established and documented in the literature so far, and we do not claim novelty on that point. Our added value is elsewhere. First, we mapped adherence across all three EMERGE/ABC phases using CMA9 in mont

---

## [Decision Letter · Decision Letter 3]

29 Dec 2025

Dear Dr. Mucherino,

Thank you for submitting your manuscript to PLOS ONE. After careful consideration, we feel that it has merit but does not fully meet PLOS ONE’s publication criteria as it currently stands. Therefore, we invite you to submit a revised version of the manuscript that addresses the points raised during the review process.

We look forward to receiving your revised manuscript.

Kind regards,

Yee Gary Ang, MBBS MPH

Academic Editor

PLOS One

Journal Requirements:

Additional Editor Comments:

We have invited multiple reviewers and 2 of them have came back with some comments.

Please address them and resubmit if applicable.

Happy New Year

Reviewers' comments:

Reviewer's Responses to Questions

**Comments to the Author**

Reviewer #1: (No Response)

Reviewer #4: (No Response)

2. Is the manuscript technically sound, and do the data support the conclusions?

Reviewer #1: Yes

Reviewer #4: Yes

3. Has the statistical analysis been performed appropriately and rigorously?

Reviewer #1: Yes

Reviewer #4: Yes

4. Have the authors made all data underlying the findings in their manuscript fully available?

Reviewer #1: Yes

Reviewer #4: Yes

5. Is the manuscript presented in an intelligible fashion and written in standard English?

Reviewer #1: Yes

Reviewer #4: Yes

Reviewer #1: I commented previously that the prescription system in Netherland should be clarified in methods section. Still, it is not clear for me. Because The prescription of medication and its practices are the foundation of the entire article, this should be described in sufficient detail in the methods section.

Otherwise, I am now satisfied with the revised text.

Reviewer #4: The authors have adequately addressed most of my comments. Here are some minor revisions that still need to be done.

• Line 81: The term ATC was added here, but is defined only on line 91. Please move the definition to the first occurrence.

• Line 113: What was the initiation date?

• Lines 128-132: The definition and calculation of implementation should be moved here, not into the statistical analysis section.

• Lines 171, 298, 299, 300, 317, 338: FU should be written in long.

• Statistical analysis (see my comment above): Much of what is written in this section should be moved earlier, when these variables are defined, notably lines 179-186 and 188-191.

• Line 207: times a day

• Line 266: (98.8%) at the end of...

• Table 2: Nr is not defined under the table and should probably be abbreviated by no. or N

• Lines 330-332: Logistic regression models and OR are not mentioned and defined in the analysis section. This information should be presented before.

• Line 332: There are still three decimal places there, and elsewhere as well.

• Table 3: aOR and 95% CI are not defined under the table, and there is a 0.477 (three decimals).

• Line 344: p<0.005 should probably be p<0.05

• Lines 363-368: The section repeats what was added in lines 350-352.

• Line 374: "better" or "greatest" should probably be removed.

• Line 473: diagnosed with type 2 diabetes?

**Do you want your identity to be public for this peer review?** For information about this choice, including consent withdrawal, please see our Privacy Policy

Reviewer #1: No

Reviewer #4: **Yes:** Line Guénette

You may also use PLOS’s free figure tool, NAAS, to help you prepare publication quality figures: https://journals.plos.org/plosone/s/figures#loc-tools-for-figure-preparation

---

## [Author Response · Author response to Decision Letter 4]

7 Jan 2026

Reviewers’ Comments

Reviewer #1 Comments

General comment: I commented previously that the prescription system in Netherland should be clarified in methods section. Still, it is not clear for me. Because The prescription of medication and its practices are the foundation of the entire article, this should be described in sufficient detail in the methods section. Otherwise, I am now satisfied with the revised text.

Response: Dear Reviewer, we understand your point. We have added further details on these aspects in the methods section, in the subsection Data source and study population selection, as follows: “This study was carried out based on data sourced from Nivel Primary Care Database (Nivel-PCD). Nivel-PCD systematically collects data from approximately 500 general practices in the Netherlands (around 10% of the Dutch population) including prescribed medications recorded by general practitioners (GPs) [18]. Hence, Nivel-PCD contains GPs’ prescribing records, namely medications ordered by the GPs within the Dutch health system. Pharmacy dispensing/claims data is not captured in Nivel-PCD. In Dutch primary care, long-term therapies are issued as repeat prescriptions with an authorisation typically valid for up to 12 months. Community pharmacies generally dispense 1-3 per month quantities per issue. Insulins and oral antidiabetic drugs (OADs) follow the same framework. Each prescription record in Nivel-PCD includes the calendar date, Anatomical Therapeutic Chemical Classification (ATC) code, prescribed daily dose/instructions, and intended duration or quantity. These fields were used to derive days of coverage per prescription and to build 12-month prescription histories from the index date (first OAD prescription). For this study, these data were integrated with patient characteristics, healthcare consultations, patient morbidity profiles, laboratory results, and the primary or secondary healthcare provider managing diabetes care.” To be more clear we also added the recent reference regarding the prescription info available within the Nivel-PCD in the Netherlands (now reference no 18): Vanhommerig JW, Verheij RA, Hek K, Ramerman L, Hooiveld M, Veldhuijzen NJ, Veldkamp R, van Dronkelaar C, Stelma FF, Knottnerus BJ, Meijer WM, Hasselaar J, Overbeek LI. Data Resource Profile: Nivel Primary Care Database (Nivel-PCD), The Netherlands. Int J Epidemiol. 2025 Feb 16;54(2):dyaf017. doi: 10.1093/ije/dyaf017.

Reviewer #4 Comments

General comment: The authors have adequately addressed most of my comments. Here are some minor revisions that still need to be done.

Response: Dear Reviewer, again, we greatly appreciate these additional specific comments on the manuscript. They’re precise and pertinent and helped to improve the clarity of the manuscript. We made all requested changes and hope you will appreciate the revised version of the resubmitted manuscript. Below you will find point-by-point responses to each of your comments. Thank you.

Comment 1: Line 81: The term ATC was added here, but is defined only on line 91. Please move the definition to the first occurrence.

Response 1: Thanks. We’ve now cited in full the term ATC in the first occurrence.

Comment 2: Line 113: What was the initiation date?

Response 2: The initiation date is the index date, which is the calendar date of the patient’s first prescription of the index OAD. This date marks the start of follow-up. As the dispensing data were unavailable, evidence of the initiation phase was operationalised as having ≥2 prescriptions within 12 months after the index date; this does not change the initiation date itself. If the patient had more than 2 following OAD prescriptions we flagged the patient as “initiator”, hence the index date corresponds to the OAD therapy initiation for this patient. We explained this in the manuscript were suggested.

Comment 3: Lines 128-132: The definition and calculation of implementation should be moved here, not into the statistical analysis section.

Response 3: We agree with this perspective. We have in fact moved the entire description and definition of the implementation phase from statistical analysis to the paragraph entitled “Trajectory modelling and medication adherence estimation”. Thank you.

Comment 4: Lines 171, 298, 299, 300, 317, 338: FU should be written in long.

Response 4: We have now always written the word "follow-up" in full.

Comment 5: Statistical analysis (see my comment above): Much of what is written in this section should be moved earlier, when these variables are defined, notably lines 179-186 and 188-191.

Response 5: As said, we agree. We’ve moved majority of these descriptions to the statistical analysis section.

Comment 6: Line 207: times a day

Response 6: Amended. Thanks.

Comment 7: Line 266: (98.8%) at the end of...

Response 7: Amended. Thanks.

Comment 8: Table 2: Nr is not defined under the table and should probably be abbreviated by no. or N

Response 8: We changed it with “N”. Thanks.

Comment 9: Lines 330-332: Logistic regression models and OR are not mentioned and defined in the analysis section. This information should be presented before.

Response 9: We added a paragraph with the pertinent specifications within the statistical analysis section, as follows: “Multivariate logistic regression was performed to identify factors associated with adherence clusters. A stepwise selection procedure was used to select covariates to be included in the models. These covariates included: sex, age category (<25, 26–39, 40–59, 60–79, ≥80 years), high-risk T2D status per NHG guidance [27], hypertension at baseline, baseline HbA1c within target (≤53 mmol/mol; 7%), baseline blood pressure within target (<140/90 mmHg), and insulin addition during follow-up (post-index indicator). Results are presented as adjusted odds ratios (aORs) with 95% confidence intervals and two-sided p-values (α=0.05).”

Comment 10: Line 332: There are still three decimal places there, and elsewhere as well.

Response 10: We have corrected and double-checked the entire paper, including the supplementary material. Thanks.

Comment 11: Table 3: aOR and 95% CI are not defined under the table, and there is a 0.477 (three decimals).

Response 11: We’ve now defined it at the bottom of the table and amended the numbers with 3 decimals. Thanks.

Comment 12: Line 344: p<0.005 should probably be p<0.05

Response 12: Yes, amended.

Comment 13: Lines 363-368: The section repeats what was added in lines 350-352.

Response 13: We merged these two paragraphs in one. Thanks.

Comment 14: Line 374: "better" or "greatest" should probably be removed.

Response 14: We’ve now reworded.

Comment 15: Line 473: diagnosed with type 2 diabetes?

Response 15: Yes, amended.

---

## [Decision Letter · Decision Letter 4]

18 Jan 2026

Medication adherence trajectories and association with risk factors and clinical outcomes in type 2 diabetes treatment

PONE-D-25-41866R4

Dear Dr. Mucherino,

We’re pleased to inform you that your manuscript has been judged scientifically suitable for publication and will be formally accepted for publication once it meets all outstanding technical requirements.

Kind regards,

Yee Gary Ang, MBBS MPH

Academic Editor

PLOS One

Additional Editor Comments (optional):

Reviewers' comments:

Reviewer's Responses to Questions

**Comments to the Author**

Reviewer #1: All comments have been addressed

Reviewer #4: All comments have been addressed

2. Is the manuscript technically sound, and do the data support the conclusions?

Reviewer #1: Yes

Reviewer #4: Yes

3. Has the statistical analysis been performed appropriately and rigorously?

Reviewer #1: Yes

Reviewer #4: Yes

4. Have the authors made all data underlying the findings in their manuscript fully available?

Reviewer #1: Yes

Reviewer #4: No

5. Is the manuscript presented in an intelligible fashion and written in standard English?

Reviewer #1: Yes

Reviewer #4: Yes

Reviewer #1: The prescription system of Netherland is now clarified sufficiently in methods section.

I am now satisfied with the revised manuscript.

Reviewer #4: (No Response)

**Do you want your identity to be public for this peer review?** For information about this choice, including consent withdrawal, please see our Privacy Policy

Reviewer #1: No

Reviewer #4: No

---

## [Editor Report · Acceptance letter]

PONE-D-25-41866R4

PLOS One

Dear Dr. Mucherino,

I'm pleased to inform you that your manuscript has been deemed suitable for publication in PLOS One. Congratulations! Your manuscript is now being handed over to our production team.

Kind regards,

on behalf of

Dr. Yee Gary Ang

Academic Editor

PLOS One